

# Concentrations, composition, and sources of ice-nucleating particles in the Canadian High Arctic during spring 2016

Meng Si[1], Erin Evoy[1], Jingwei Yun[1], Yu Xi[1], Sarah Hanna[1], Alina Chivulescu[2], Kevin Rawlings[2], Andrew Platt[2], Daniel Kunkel[3], Peter Hoor[3], Sangeeta Sharma[2], W. Richard Leaitch[2], Allan K. Bertram[1]

[1]Department of Chemistry, University of British Columbia, Vancouver, BC, V6T 1Z1, Canada
[2]Climate Research Divisions, Environment and Climate Change Canada, Toronto, ON, M3H 5T4, Canada
[3]Institute for Atmospheric Physics, Johannes Gutenberg University Mainz, Mainz, 55128, Germany

*Correspondence to*: Allan K. Bertram (*bertram@chem.ubc.ca*)

**Abstract.** Modelling studies suggest that the climate and the hydrological cycle are sensitive to the concentrations of ice-
nucleating particles (INPs). However, the concentrations, composition, and sources of INPs in the atmosphere remain uncertain. Here we report daily concentrations of INPs and tracers of mineral dust (Al, Fe, Ti, and Mn), sea spray aerosol ($Na^+$ and $Cl^-$), and anthropogenic aerosol (Zn, Pb, $NO_3^-$, $NH_4^+$, and non-sea-salt $SO_4^{2-}$) at Alert, Canada during a three-week campaign in March 2016. The average INP concentrations measured in the immersion freezing mode were approximately $0.005 \pm 0.002$ L$^{-1}$, $0.020 \pm 0.004$ L$^{-1}$, and $0.186 \pm 0.040$ L$^{-1}$ at -15 ºC, -20 ºC, and -25 ºC, respectively. These concentrations
are within the range of concentrations measured previously in the Arctic at ground level or sea level. Mineral dust tracers all correlated with INPs at -25 ºC (correlation coefficient, $R$, ranged from 0.70 to 0.76), suggesting that mineral dust was a major contributor to the INP population. Particle dispersion modelling suggests that the source of the mineral dust may have been the long-range transported dust from the Gobi desert. Sea spray tracers were anti-correlated with INPs at -25 ºC ($R$ = -0.56). In addition, INP concentrations at -25 ºC divided by mass concentrations of aluminum were anti-correlated with sea
spray tracers ($R$ = -0.51 and -0.55 for $Na^+$ and $Cl^-$, respectively), suggesting that the components of sea spray aerosol suppressed the ice-nucleating ability of mineral dust in the immersion freezing mode. Correlations between INPs and anthropogenic aerosol tracers were not statistically significant. These results will improve our understanding of INPs in the Arctic during spring.

## 1 Introduction

The formation of ice in clouds can be initiated by homogeneous or heterogeneous nucleation. Heterogeneous nucleation of ice in clouds occurs on only a subset of atmospheric particles, referred to as ice-nucleating particles (INPs) (Vali et al., 2015). A variety of aerosol particle types have been identified as possible INPs, including, but not limited to, mineral dust, sea spray aerosol containing biological material, and primary biological particles from terrestrial sources (Coluzza et al., 2017; Cziczo et al., 2017; Hoose and Möhler, 2012; Kanji et al., 2017; Murray et al., 2012). INPs can change the frequency
and properties of clouds in the atmosphere and influence climate and precipitation (Andreae and Rosenfeld, 2008; Curry,





1995; DeMott et al., 2010; Du et al., 2011; Lohmann and Feichter, 2005; Prenni et al., 2007; Xie et al., 2013). As a result, to predict precipitation and Earth's climate, an understanding of the concentrations, composition, and sources of INPs in the atmosphere is required.

INPs have been measured in the Arctic since as early as the 1970s (Bigg, 1996; Bigg and Leck, 2001; Borys, 1989; Conen et
al., 2016; Creamean et al., 2018; DeMott et al., 2016; Flyger et al., 1973, 1976; Flyger and Heidam, 1978; Fountain and Ohtake, 1985; Mason et al., 2016; McFarquhar et al., 2011; Prenni et al., 2007, 2009; Radke et al., 1976; Rogers et al., 2001). Nevertheless, our understanding of the concentrations, composition, and sources of INPs in this region is incomplete. In the following, we focus on INPs in the Arctic during the spring. In the Arctic winter-spring, land is mostly covered by snow and hence local emissions of mineral dust and primary terrestrial biological particles are expected to be small. In
addition, a large fraction of the Arctic Ocean is covered by ice, limiting emissions of sea spray aerosol. On the other hand, during the winter-spring, long-range transport of particles from mid-latitudes is important, and removal processes of aerosols are reduced, leading to elevated aerosol particle concentrations in the region, referred to as Arctic haze (Barrie, 1986; Barrie et al., 1981; Norman et al., 1999; Pacyna, 1995; Quinn et al., 2007; Shaw, 1995).

There have only been a small number of measurements of INPs in the Arctic during the spring (Borys, 1989; Creamean et
al., 2018; Fountain and Ohtake, 1985; Mason et al., 2016; McFarquhar et al., 2011; Radke et al., 1976; Rogers et al., 2001). Of note, Borys (1989) measured INP concentrations in April 1986 and found that low concentrations of INPs were associated with tracers of pollution, while high concentrations of INPs were associated with a tracer of mineral dust. Fountain and Ohtake (1985) measured surface INP concentrations in Alaska from August 1978 to March 1979 and observed that high INP concentrations in March were correlated with air masses transported from Eurasia. Rogers et al. (2001)
measured concentrations and the chemical composition of INPs during aircraft studies in May 1998. Chemical analysis of the INPs indicated that a significant fraction of the INPs (approximately 37 %) contained mineral dust. Mason et al. (2016) carried out size-resolved measurements of INPs from the end of March to late July 2014. They found that a large fraction (> 60% at -20 ºC) of the INPs were larger than 1 μm, indicating supermicron particles such as mineral dust or sea spray aerosol containing biological material contributed to a majority of the INPs.

In the following, we investigate the composition and sources of INPs at Alert, Nunavut in the Canadian High Arctic from 11 to 29 March 2016. This study was carried out as part of the Network on Climate and Aerosols: Addressing Key Uncertainties in Remote Canadian Environments (NETCARE). Specifically, we measured daily the concentrations of INPs in the immersion freezing mode, and concentrations of tracers of mineral dust (Al, Fe, Ti, and Mn), sea spray aerosol ($Na^+$ and $Cl^-$), and anthropogenic aerosol (Zn, Pb, $NO_3^-$, $NH_4^+$, and non-sea-salt $SO_4^{2-}$). These data were used to determine if mineral dust,
sea spray aerosol, and anthropogenic aerosol are a major contributor to the INP population in the Canadian High Arctic during spring. There have only been a very few studies that have investigated the importance of sea spray aerosol and anthropogenic aerosol to the INP population in the Arctic during spring. Although a few studies have identified mineral dust particles as an important contributor to the INP population in the Arctic during spring, additional studies are still needed to





determine how often mineral dust is an important contributor. The measurements together with particle dispersion modelling were also used to assess the source of the INPs.

## 2 Methods

### 2.1 Sampling location

Sampling was conducted at the Dr. Neil Trivett Global Atmosphere Watch Observatory (82.5º N, 62.5º W) at Alert, Nunavut (Fig. 1), a research laboratory operated by Environment and Climate Change Canada. The research laboratory is on a plateau that is approximately 185 m above sea level and is powered by electricity generated by the Canadian Forces Station Alert, which is 6 km to the north of the research laboratory. The population at Alert is about 75 on a regular basis. The closest community is Grise Fiord (population 129) located approximately 800 km to the south of Alert.

**2.2 INP measurements**

Concentrations of INPs were determined by first collecting particles on hydrophobic glass slides with an inertia impactor followed by determining the freezing temperatures of the collected aerosol particles with the droplet freezing technique (DeMott et al., 2016; Irish et al., 2018). Details are given below.

### 2.2.1 Inertia impactor

The inertia impactor (model 100-180nm-10lmp; MSP Corp., Shoreview, MN, USA) consisted of two impactor stages. The first impactor stage collected particles with aerodynamic diameters > 10 µm, and the second impactor stage collected particles with aerodynamic diameters between 0.18 and 10 µm. Three circular hydrophobic glass slides (HR3-277; Hampton Research, USA) were placed on the second impactor stage to collect particles for INP analysis. Due to the design of the impactor, the collected particles were concentrated into 300 spots on the hydrophobic glass slides. Particles collected on the

first stage were not analyzed. The hydrophobic glass slides were cleaned with Millipore water and dried with ultrapure nitrogen gas before being sent to the field for particle collection. After collection, all slides were placed in petri dishes, wrapped in aluminum foil, and stored at 4 ºC before analysis. The inertia impactor was operated at a flow rate of 10 L min$^{-1}$ and the average collection time for INP samples was approximately 2h.

The inertia impactor was located within the Dr. Neil Trivett Global Atmosphere Watch Observatory. Aerosol particles were

first sampled through a louvered total suspended particulate (TSP) inlet (Mesa Labs Inc., Butler, NJ, USA) that was approximately 10 m above ground level. Next, the aerosol particles were passed through a humidifier (model FC125-240; Perma Pure LLC, Lakewood, NJ, USA) to condition the aerosol particles to an average relative humidity (RH) of 84 % at room temperature. Finally, the aerosol particles were passed through the inertia impactor to collect the particles on the hydrophobic glass slides for INP analysis.



When sampling aerosol particles with an inertia impactor, a possible issue is particle rebound from the collection substrate. Rebound occurs when the kinetic energy of the particles striking the impactor substrate exceeds the adhesion and dissipation energies at impact (Bateman et al., 2014). If rebound is a factor, the measured INP concentrations will be lower than the actual INP concentrations in the atmosphere. Previous work has shown that particle rebound can be reduced when RH is

above 70 % (Bateman et al., 2014; Chen et al., 2011; Fang et al., 1991), although this will depend on the chemical composition of the particles. In addition, field measurements of INP concentrations using a similar inertia impactor have shown good agreement with INP concentrations measured by a continuous flow diffusion chamber (a technique that is not susceptible to rebound) when the RH of the sampled aerosol stream was as low as 40 % (DeMott et al., 2017; Mason et al., 2015). In our experiments, the RH was increased to an average value of 84 % at room temperature with a humidifier to

reduce rebound. However, rebound cannot be completely ruled out.

### 2.2.2 Droplet freezing technique

The INP number concentrations in the immersion freezing mode were determined with the droplet freezing technique (Iannone et al., 2011; Mason et al., 2015; Wheeler et al., 2015). Briefly, the hydrophobic glass slides used to collect aerosol particles with the inertia impactor were placed in a temperature- and humidity-controlled flow cell coupled to an optical

microscope (Axiolab; Zeiss, Oberkochen, Germany) with a CCD camera. Typically between 15-25 spots of particles (out of 300 spots generated by the inertia impactor) were visible in the microscopic field of view. After locating the hydrophobic glass slides in the flow cell, the temperature in the flow cell was decreased to approximately 0 ºC and the RH was increased to above water saturation, resulting in water droplets with diameters of approximately 100-500 μm condensing on the spots of particles as well as other areas on the glass slides. Droplets that condensed on the spots of particles are referred to as spot

droplets, while droplets that condensed on other areas of the slide are referred to as non-spot droplets. After condensation of the water droplets, the flow cell was cooled down to -40 ºC at a rate of -10 ºC min$^{-1}$ while images of the droplets were recorded. The freezing temperature of each droplet was then determined from the images. The droplets that contain spots of particles were also identified from these images. The number of INPs within the microscopic field of view was calculated as a function of temperature using the following equation:

$$\#INP(T) = \left( -ln \left( \frac{N_{us}(T)}{N_s} \right) \right) N_s \,, \tag{1}$$

where $N_{us}(T)$ is the number of unfrozen spot droplets at temperature $T$, and $N_s$ is the total number of spots analyzed in the microscopic field of view. Equation (1) accounts for the possibility of multiple INPs in a spot droplet (Vali, 1971).

Equation (1) assumes that each droplet covered only one spot. However, sometimes more than one droplet formed on one spot. In these cases, the first droplet to freeze was considered in Eq. (1), which should give the equivalent result to one

droplet condensing on one spot. Another situation is when one droplet covered two spots (this occurred for less than 5 % of the total analyzed spot droplets). For these cases, an upper limit of the number of INPs was calculated by assuming two droplets covered the two spots and both droplets froze at the observed freezing temperature. A lower limit was calculated by



assuming two droplets covered the two spots with one droplet freezing at the observed freezing temperature, and the other freezing at the homogeneous freezing temperatures of the droplets. A similar approach was applied to cases where one droplet covered three or more spots.

Freezing of non-spot droplets was less frequent than freezing of spot droplets at temperatures ≥ -25 °C. For example, the ratio of frozen non-spot droplets to frozen spot droplets was 0.2 at -25 °C. Freezing of non-spot droplets may have been due to INPs < 0.18 μm in diameter not focused into the spots. To take into account the INPs in the non-spot droplets, we assumed each frozen non-spot droplet contained one INP, and the number of frozen non-spot droplets was added to Eq. (1).

During the freezing experiments, most freezing events occurred by immersion freezing, while some occurred by contact freezing, which refers to the freezing of liquid droplets coming into contact with neighboring frozen droplets. Contact freezing only accounted for approximately 2 % of the total freezing events, and droplets that froze by contact freezing were not considered when determining the INP concentrations.

The INP concentration as a function of temperature in the atmosphere, $[INPs(T)]$, was calculated using the following equation:

$$[INPs(T)] = \#INP(T)\left(\frac{300}{N_s V}\right), \tag{2}$$

where the ratio of $300/N_s$ accounts for the fact that only a fraction of total spots (300 spots in total) was analyzed with the droplet freezing technique and $V$ is the total volume of air sampled by the inertia impactor. Uncertainties in $[INPs(T)]$ due to the limited number of freezing events detected were determined using the method given in Koop et al. (1997), and represent the 95 % confidence interval.

Droplet freezing experiments were also performed on glass slides that were not exposed to any particles, referred to as blanks. In these cases, we assumed the $\#INP(T)$ was equal to the number of observed freezing events since multiple INPs within the same droplet was unlikely at temperatures ≥ -25 °C. $[INPs(T)]$ was then calculated using Eq. (2) with the assumptions that $N_s$ is 21 (average number of spots analyzed in one experiment) and $V$ is 1208 L (average air volume sampled).

### 2.3 Meteorological parameters

Local ground-level meteorological conditions were monitored at the site by Environment and Climate Change Canada. The March 2016 data was retrieved from climate.weather.gc.ca (climate IDs 2400305 and 2400306). The ambient temperature, ambient RH, wind speed, and wind direction were measured hourly. The sum of the total rainfall and the water equivalent of the total snowfall in millimeters were measured daily as total precipitation.

### 2.4 Tracers of mineral dust, sea spray, and anthropogenic aerosol

The elements Al, Fe, Ti, and Mn were used as tracers of mineral dust, as done previously (Balasubramanian et al., 2003; Barrie and Barrie, 1990; Formenti et al., 2003; Malm et al., 1994; Quinn et al., 2004). These elements are components of the




Earth's crust (Usher et al., 2003; Wedepohl, 1995). The species $Na^+$ and $Cl^-$, which are the major inorganic components of seawater (Holland, 1978), were used as tracers of sea spray, as done previously (Balasubramanian et al., 2003; Malm et al., 1994; Quinn et al., 2002, 2004). For tracers of anthropogenic aerosol, we used Zn, Pb, $NO_3^-$, $NH_4^+$, and non-sea-salt $SO_4^{2-}$ (nss-$SO_4^{2-}$). Pb and Zn are almost exclusively from anthropogenic sources (Macdonald et al., 2000; Nriagu and Pacyna, 1988;

Pacyna, 1995). The major anthropogenic sources for Pb are gasoline combustion and, to a lesser extent, non-ferrous metal industry and fossil fuel combustion, and the major anthropogenic sources for Zn are non-ferrous metal industry followed by fossil fuel combustion (Barrie et al., 1992; Nriagu and Pacyna, 1988; Pacyna, 1995). $NO_3^-$, $NH_4^+$, and nss-$SO_4^{2-}$ can come from both anthropogenic and natural sources, but mostly from anthropogenic sources. $NO_3^-$ is mainly formed from $NO_x$, which is emitted from combustion processes (Seinfeld and Pandis, 2006). $NH_4^+$ originates mainly from agricultural activities

(Follett and Hatfield, 2001). The main anthropogenic source of nss-$SO_4^{2-}$ is fossil fuel combustion (Schwikowski et al., 1999; Ward, 2009).

To determine the concentrations of the tracers discussed above, aerosol particulate samples were collected on 20 x 25 cm Whatman-41 quartz filters daily using a high-volume sampler (Barrie et al., 1981, 1989), which was located approximately 500 m away from the laboratory on the ground. The face velocity of sampling (50 cm $s^{-1}$) and typical filter loadings ensured

collection efficiencies better than 95 % (Watts et al., 1987). The average sampling time was 23.5 h except for the first sample, which was collected over two days due to a storm making it difficult to change the filter. The average total air volume was roughly 2,300 $m^3$ at standard conditions of 1 atm pressure and 0 ºC. The precision of volume sampled was estimated to be ± 5 % (Sirois and Barrie, 1999). The quartz filter samples were stored at room temperature before analysis.

The concentrations of Al, Fe, Ti, Mn, Zn, and Pb were determined using inductively coupled plasma atomic emission

spectroscopy (ICP-AES). These experiments were carried out at Chester LabNet, Oregon, USA. Punches from the quartz filters were submerged in a solution containing ultrapure $HNO_3$ and HCl. The solution was then heated and sonicated, and then further diluted and filtered before being nebulized and analyzed by the ICP-AES (Perkin-Elmer Optima 8300). Sample duplicates were also analyzed to estimate the method precision. The accuracy and precision of the technique were estimated to be ± 10 %.

The concentrations of $Cl^-$, $Na^+$, $NO_3^-$, $SO_4^{2-}$, and $NH_4^+$ were determined using ion chromatography (IC) (Macdonald et al., 2017; Toom-Sauntry and Barrie, 2002). These experiments were carried out at Environment and Climate Change Canada in Ontario, Canada. To quantify water-soluble cations and anions, punches taken from each quartz filter were extracted in deionized water, and the extraction solution was passed through an IC (Dionex IC: DX600). The concentration of nss-$SO_4^{2-}$ was calculated using the following equation based on the assumption that the chemical composition of sea salt particles is

the same as that of seawater (Millero, 1974):

$$[nss - SO_4^{2-}] = [SO_4^{2-}] - 0.14[Cl^-] , \qquad (3)$$




**2.5 Particle dispersion modelling**

The source regions of measured air masses were investigated using the Lagrangian particle dispersion model FLEXPART (Stohl et al., 2005), which was driven using operational meteorological analyses from the European Center for Medium-Range Weather Forecasts (ECMWF). FLEXPART was run at hourly intervals in the backward mode for each sample collected for INP analysis. In each run, 40,000 particles were released over 1 minute in a 0.1 deg x 0.1 deg area. The particles were followed backward for 20 days with output generated at 1-hour intervals. In backward mode FLEXPART provides potential emission sensitivities as output. This is the response function of a source-receptor matrix (Seibert and Frank, 2004) and corresponds directly to the residence time of the released particles in a given volume of air. For each run, the hourly output was integrated over the 20 days to produce a potential emission sensitivity (PES) plot. For a given INP sample, the mean PES plot was generated by averaging all hourly PES plots. Since the focus of this study are INP sources close to the surface, near surface PES plots (from the surface up to 100 m) were plotted as footprint PES plots.

**2.6 Statistical analysis**

To compute a correlation coefficient ($R$), Pearson correlation analysis was applied between INPs and the variables measured in this study. $P$-values were also calculated using t-test for Student's t-distribution to determine if the correlations were statistically significant at the 95 % confidence level ($P < 0.05$).

**3 Results and discussion**

**3.1 Concentrations of INPs**

Concentrations of INPs in the immersion mode are plotted as a function of temperature in Fig. 2. Concentrations of INPs measured were higher than concentrations of the blanks. In addition, all of the samples have warmer onset temperatures than the blanks, with onset temperatures of the samples varying from approximately -11 to -22 ℃. For the remainder of this document, we focus on INP concentrations at -15, -20, and -25 ℃. INP concentrations at temperatures warmer than -15 ℃ are not discussed because freezing events at these temperatures were rare. INP concentrations at temperatures below -25 ℃ are not discussed since freezing of the blanks became significant at these temperatures.

In the current study, the mean number concentration of INPs was $0.005 \pm 0.002$ L$^{-1}$ at -15 ℃, $0.020 \pm 0.004$ L$^{-1}$ at -20 ℃, and $0.186 \pm 0.040$ L$^{-1}$ at -25 ℃ (Fig. 3). These concentrations are within the range of INP concentrations measured in previous studies at ground level or sea level in the Arctic (Fig. 3). The sampling platform, location, and dates of previous Arctic INP studies at ground level or sea level that are shown in Fig. 3 are summarized in Table 1 and Fig. 1 for comparison purposes.




The time series of INP concentrations measured in the current study at -15, -20, and -25 ºC is plotted in Fig. 4. The difference between the highest INP concentration at -25 ºC (measured on March 28) and the lowest INP concentration at -25 ºC (measured on March 17) was roughly a factor of 50.

### 3.2 Meteorological parameters and correlations with INPs

Shown in Fig. 5 is the time series of meteorological parameters measured at Alert during the field campaign. Precipitation was rare throughout the campaign, and the average ambient temperature was approximately -30 ºC. The RH was above 70 % for most of the time. The wind speed was below 10 km h$^{-1}$ except for two storm events during the first part of the campaign. During the field campaign, the wind came mainly from the SW (more than half of the time) with some contribution from the SE and NW.

In the current study, INP concentrations at freezing temperatures of -15, -20, and -25 ºC were not correlated with any meteorological parameters (Table S1). This result is consistent with the results from Fountain and Ohtake (1985), who also did not find any correlations between meteorological variables (including air temperature, precipitation, fog, and wind direction, etc.) and INP concentrations measured at Barrow, Alaska from August 1978 to April 1979. In contrast, measurements by Radke et al. (1976) in Barrow, Alaska during March found that the INP concentrations were affected by local weather conditions.

### 3.3 Tracers of mineral dust: concentrations, correlations with INPs, and sources

The time series of concentrations of mineral dust tracers (Al, Fe, Ti, and Mn) is plotted in Fig. 6. Figure 6 shows that concentrations of different tracers of mineral dust are correlated with each other. Shown in Table 2 is a correlation coefficient matrix between each of the tracers based on a Pearson correlation analysis. Correlations that are both strong ($R \geq$ 0.7) and statistically significant ($P < 0.05$) are highlighted in bold and red. The correlation coefficients show that the mineral dust tracers are strongly correlated with each other ($R \geq 0.85$), which is expected since these elements are almost exclusively from the mineral dust sources.

In Fig. 7(a), we compared the mean mass concentration of aluminum we measured at the site during the campaign with the concentrations reported in Sirois and Barrie (1999) for the same site and for time period from 1980 through 1995. The mean mass concentration of aluminum measured during the current study was lower than the previous concentrations measured in March at Alert, indicating that the concentration of mineral dust during the current campaign was lower than many of the previous measurements at Alert during March.

Pearson regression analysis was also done between INP concentrations at -15, -20 and -25 ºC, and tracers of mineral dust. The correlation coefficients and $P$-values are summarized in Table 3. The linear regression plots between INP concentrations at -25 ºC and mineral dust tracers are shown in Fig. 8. When calculating the correlation coefficients, a value of zero was used when tracer concentrations and INP concentrations were below the detection limits. It is important to keep in mind that the collection time was different for INPs (~2h) and the aerosol tracers (~24h), and that the analysis presented here is based on



the assumption that the average concentrations of the aerosol tracers during the INP sampling time (~2h) were the same as the average concentrations measured by the high-volume sampler (~24h). As shown in Table 3, at -15 ºC, the correlations were not statistically significant ($P > 0.05$). At -20 ºC, the correlations between INPs and Ti and Mn were not statistically significant. Al and Fe were moderately correlated with INPs at -20 ºC ($R = 0.53$ and $0.60$, respectively), and the correlations

were statistically significant ($P < 0.05$). At -25 ºC, all four tracers were strongly correlated with INPs ($R$ ranged from 0.70 to 0.76), and the correlations were statistically significant. This suggests that mineral dust was a component of the sampled INPs at a freezing temperature of -25 ºC. This is consistent with previous field measurements that have identified mineral dust as a major component of the INP population at different locations (Boose et al., 2016; Cziczo et al., 2013; DeMott et al., 2003; Klein et al., 2010; Pratt et al., 2009; Prenni et al., 2009). A previous study of ice nucleation of Arctic aerosol found

that the elements of crustal or natural dust were associated with high concentrations of INPs at -15 ºC and -25 ºC (Borys, 1989). Chemical analysis of the INPs during aircraft studies in the Arctic indicated that a significant fraction of the INPs (approximately 37 %) contained mineral dust (Rogers et al., 2001). Another study of snow crystals during summer on the Greenland Ice Cap also showed that the natural snow crystals mainly formed on clay mineral particles by heterogeneous nucleation (Kumai and Francis, 1962).

Mineral dust at Alert can come from both local sources and long-range transport, but during the spring local sources are not likely important since land is covered with snow during this time of the year. To investigate the origins of mineral dust measured in this study, the FLEXPART model was used to generate the footprint PES plots, which show the residence time of aerosol particles in the layer from 0 to 100 m in altitude during the 20 days prior to sampling. The results are shown in Fig. 9. Figure 9(A) shows the surface coverage type on the first day of sampling (March 12). The sampling location at Alert was

surrounded by ice and snow during the campaign. Figure 9(B) shows the average footprint PES for all samples, and Fig. 9(C) shows the average footprint PES for the four samples with the highest mineral dust concentrations, which were collected on March 21, 22, 27, and 28. Shown in Fig. 9(D) is the ratio of Fig. 9(C) to Fig. 9(B), which is often used as a sensitive method to identify the source regions of a component under investigation (Hirdman et al., 2010). In these types of plots, a value close to one indicates a more likely source region. Figure 9(D) suggests that the North Pacific Ocean, Alaska, and the Gobi

Desert were possible source regions of the mineral dust sampled. Alaska was covered in snow during the sampling period, so this was not likely the source of the mineral dust, although mineral dust re-suspended from blowing snow cannot be ruled out. Mineral dust from shore lines also can not be ruled out. The North Pacific Ocean is not likely the source of mineral dust either. Based on the surface coverage types (Fig. 9(A)) and the ratio plot (Fig. 9(D)), a possible source of the mineral dust was long-range transport of dust from the Gobi desert. This conclusion is consistent with previous studies that have shown

that a substantial fraction of the dust reaching Alert in spring months comes from long-range transport of Asian dust (Drab et al., 2002; Franzén et al., 1994; Pacyna and Ottar, 1989; Sirois and Barrie, 1999; Welch et al., 1991).





### 3.4 Tracers of sea spray aerosol: concentrations and correlations with INPs

The time series of sea spray aerosol tracers ($Na^+$ and $Cl^-$) is shown in Fig. 10, and the results of the Pearson correlation analysis between these tracers are listed in Table 2. The correlation coefficients show that sea spray tracers are strongly correlated with each other ($R = 0.95$), which is expected since these species are almost exclusively from sea spray.

A comparison between the mean mass concentrations of $Na^+$ and $Cl^-$ measured in the current study and the values measured by Sirois and Barrie (1999) from 1980 to 1995 is shown in Fig. 7(b-c). Figure 7(b-c) illustrates that the concentrations of $Na^+$ and $Cl^-$ measured in the current study were consistent with many of previous measurements at Alert during March.

The correlation coefficients between INP concentrations at -15, -20, and -25 °C, and tracers of sea spray aerosol are listed in Table 3. The liner regression plots between INP concentrations at -25 °C and the sea spray tracers are shown in Fig. 11. At -

15 °C and -20 °C, the correlations between INP concentrations and tracers of sea spray aerosol were not statistically significant ($P > 0.05$). At -25 °C, the sea spray tracers were negatively correlated with INPs with moderate correlation coefficients ($R = -0.56$) and statistical significance ($P < 0.05$). Previous field studies and modelling studies have suggested sea spray aerosol as an important source of ambient INPs in marine environments when other sources of INPs, such as mineral dust, are low (Burrows et al., 2013; DeMott et al., 2016; Rosinski et al., 1986, 1988; Schnell, 1982; Vergara-

Temprado et al., 2017; Wilson et al., 2015). Our results suggest that mineral dust is a more important source of INPs than sea spray aerosol for the time and location studied.

Due to the way the freezing experiments was conducted, any particles that were externally mixed prior to the collection will be internally mixed during the freezing process. As a result, the droplets formed during the freezing experiments contained salts. Reischel and Vali (1975) studied the effects of 0.01M, 0.1M and 1M solutions of NaCl on the nucleating ability of

kaolin, and found that the presence of NaCl led to lower freezing temperatures, by as much as 4 °C, for kaolin. A recent study by Whale et al. (2018) also found that a 0.015 M NaCl solution caused a decrease in freezing temperature of approximately 2 to 4 °C for BCS376 microcline, Eifel sanidine, quartz and Arizona test dust, but had no effect on silica. To investigate the influence of NaCl on the ice-nucleating ability of mineral dust at -25 °C in the current study, INP concentrations at a freezing temperature of -25 °C ($[INPs]_{-25°C}$) was divided by the mass concentration of aluminum ($[Al]$),

and this ratio was then plotted as a function of $Cl^-$ and $Na^+$ (Fig. 12). The variable $[INPs]_{-25°C}/[Al]$ is used as an estimation of the ice-nucleating ability of mineral dust, and it was negatively correlated with $Cl^-$ and $Na^+$ with moderate correlation coefficients ($R = -0.51$ and $-0.55$, respectively) and statistical significance ($P < 0.05$), which suggests that NaCl suppressed the ice-nucleating ability of mineral dust in the current study. This result is consistent with the two previous laboratory studies mentioned above and is the first field study that we are aware of that suggests that NaCl can suppress the ice-

nucleating ability of mineral dust particles in the immersion freezing mode even in dilute solution droplets.

Another possible explanation for the negative correlation between INP concentrations at -25 °C and sea spray tracers is that the air masses containing sea spray aerosol have relatively few mineral dust particles. In this case, the sea spray aerosol does not influence the ice-nucleating ability of mineral dust, rather appears as a negative correlation by coincidence. However,



Table 2 shows that the correlations between tracers of mineral dust and tracers of sea spray aerosol are not statistically significant, suggesting that this possible explanation is not the major reason for the negative correlation between INP concentrations at -25 ºC and sea spray tracers.

**3.5 Tracers of anthropogenic aerosol: concentrations and correlations with INPs**

The time series of anthropogenic aerosol tracers (Zn, Pb, $NO_3^-$, $NH_4^+$, and nss-$SO_4^{2-}$) is plotted in Fig. 13. The results of the Pearson correlation analysis between these different tracers are listed in Table 2. Not all the anthropogenic aerosol tracers are correlated with each other. This is not surprising since these tracers come from different sources. However, there is a correlation between Zn and Pb, and between nss-$SO_4^{2-}$ and $NH_4^+$. Zn and Pb are strongly correlated with each other ($R = 0.84$, $P < 0.01$), and nss-$SO_4^{2-}$ and $NH_4^+$ are strongly correlated with each other ($R = 0.93$, $P < 0.01$).

The comparison between the mean mass concentrations of Zn, Pb, $NO_3^-$, $SO_4^{2-}$, and $NH_4^+$ measured in the current study and the values measured by Sirois and Barrie (1999) from 1980 to 1995 is shown in Fig. 7(d-h). The concentrations of Zn, Pb, $SO_4^{2-}$, and $NH_4^+$ were lower than previous concentrations measured in March at Alert, which might be due to the reduced emissions from Eurasia after the dissolution of the former USSR in 1991 (Christensen, 1997). The concentration of $NO_3^-$ was higher than previous concentrations measured in March. An increasing trend of annual concentrations of $NO_3^-$ has been observed by others in the Arctic (Hole et al., 2006; Neftel et al., 1985).

The correlation coefficients between INP concentrations at -15, -20, and -25 ºC, and tracers of anthropogenic aerosol are listed in Table 3. None of the correlations were statistically significant at all three temperatures. In contrast, Borys (1989) found that pollution derived Arctic haze aerosol had lower INP concentrations than unpolluted troposphere aerosol.

There have been a few studies that investigated the effect of sulfate coating on the ice-nucleating properties of mineral dust

particles. Most laboratory studies have shown decreased freezing abilities for sulfate coated mineral dust particles compared to uncoated particles (Chernoff and Bertram, 2010; Eastwood et al., 2008; Gallavardin et al., 2008; Sullivan et al., 2010). However, other studies have shown an increase in the freezing temperature of droplets containing mineral dust particles in the presence of ammonium sulfate (Reischel and Vali, 1975; Whale et al., 2018). To investigate the effect of ammonium, sulfate, and the ammonium-to-sulfate ratio on the ice-nucleating ability of mineral dust at -25 ºC, the parameter [INPs]$_{-25ºC}$/[Al]

was plotted as a function of $NH_4^+$, nss-$SO_4^{2-}$, and $NH_4^+$/nss-$SO_4^{2-}$ ratio as shown in Fig. 14. The $NH_4^+$/nss-$SO_4^{2-}$ ratio was between 0 and 1. The correlations between the freezing ability of mineral dust at -25 ºC and $NH_4^+$, nss-$SO_4^{2-}$, and $NH_4^+$/nss-$SO_4^{2-}$ ratio were not statistically significant.

**4 Conclusions**

The INP concentrations measured at Alert during March 2016 fell into the range of previously reported INP concentrations

measured in the Arctic at ground level or sea level. At -25 ºC, the INP concentration was strongly correlated with tracers of mineral dust (Al, Fe, Ti, and Mn), anti-correlated with tracers of sea spray (Cl$^-$, Na$^+$), and not correlated with tracers of




anthropogenic aerosol (Zn, Pb, $NO_3^-$, $NH_4^+$, and nss-$SO_4^{2-}$) or meteorological variables. This suggests that mineral dust was a major contributor to INP populations at -25 ºC at this site during the sampling period. The ice-nucleating ability of mineral dust, represented as the ratio of INP number concentration to the mass concentration of aluminum, was also anti-correlated with the tracers of sea spray at -25 ºC, which suggests that NaCl suppressed the ice-nucleating ability of mineral dust

particles in the immersion freezing mode. This is the first field study that we are aware of that showed the suppression effect of NaCl on the ice-nucleating ability of mineral dust. The results from this study should be useful for testing and improving models used to predict INPs and climate in the Arctic.

The particle dispersion model analysis suggests that a likely source of mineral dust that caused freezing at -25 ºC was long-range transport of dust from the Gobi desert. Additional measurements of the composition of individual INP or ice crystal

residuals in the Arctic are needed to confirm the conclusions reached in the current study.

*Acknowledgement* The authors would like to thank Desiree Toom for assistance with ion chromatography analysis. We also want to acknowledge that the FLEXPART was downloaded from https://www.flexpart.eu/ and the ECMWF data were retrieved from the mars server.

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



**Table 1: Measurements of the concentrations of INPs at ground level or sea level in the Arctic. Included is the information on the sampling platform (ground vs. ship based), location, and dates.**

| Study | Platform | Location | Dates |
|---|---|---|---|
| Radke et al. (1976) | Ground | Barrow, Alaska, USA | Mar 1970 |
| Flyger & Heidam (1978) | Ground | North Greenland | Jun-Aug 1974 |
| Fountain & Ohtake (1985) | Ground | Barrow, Alaska, USA | Aug 1978-Apr 1979 |
| Bigg (1996) | Ship | Central Arctic Ocean | Aug-Oct 1991 |
| Bigg & Leck (2001) | Ship | Arctic Ocean | Jul-Sep 1996 |
| DeMott et al., (2016) | Ship | Canadian Arctic | Jul 2014 |
| Mason et al. (2016) | Ground | Alert, Nunavut, Canada | Mar-Jul 2014 |
| Conen et al. (2016) | Ground | Northern Norway | Jul 2015 |
| Creamean et al. (2018) | Ground | Alaska oilfields, USA | Mar-May 2017 |
| Current study | Ground | Alert, Nunavut, Canada | Mar 2016 |



**Table 2. The correlation coefficient ($R$) matrix between each of the measured aerosol constituents. Correlations that are statistically significant at the 95 % confidence level ($P < 0.05$) are highlighted in red. Strong correlations ($R \geq 0.7$) are highlighted in bold.**

|  | Al | Fe | Ti | Mn | Cl$^-$ | Na$^+$ | Zn | Pb | NO$_3^-$ | NH$_4^+$ |
|---|---|---|---|---|---|---|---|---|---|---|
| Fe | **0.97** | | | | | | | | | |
| Ti | **0.91** | **0.85** | | | | | | | | |
| Mn | **0.90** | **0.90** | **0.95** | | | | | | | |
| Cl$^-$ | -0.23 | -0.33 | -0.29 | -0.42 | | | | | | |
| Na$^+$ | -0.15 | -0.25 | -0.13 | -0.25 | **0.95** | | | | | |
| Zn | 0.38 | 0.49 | 0.23 | 0.38 | -0.18 | -0.15 | | | | |
| Pb | 0.46 | 0.59 | 0.33 | 0.53 | -0.39 | -0.32 | **0.84** | | | |
| NO$_3^-$ | 0.54 | 0.37 | 0.65 | 0.47 | 0.10 | 0.27 | -0.11 | -0.18 | | |
| NH$_4^+$ | 0.60 | 0.56 | **0.82** | **0.83** | -0.30 | -0.04 | 0.17 | 0.30 | 0.59 | |
| nss-SO$_4^{2-}$ | 0.41 | 0.44 | 0.65 | **0.74** | -0.46 | -0.20 | 0.20 | 0.39 | 0.33 | **0.93** |



**Table 3. Results of Pearson correlation analysis between the INP number concentrations (at freezing temperatures of -15, -20, and -25 ℃) and tracers of aerosol components measured in this study. $R$ is the correlation coefficient, $P$ is the probability value (two tailed), and the sample number is 16. Correlations that are statistically significant at the 95% confidence interval ($P < 0.05$) are highlighted in red. Strong correlations ($R \geq 0.7$) are highlighted in bold.**

| | INP number concentrations ($L^{-1}$) | | | | | |
| --- | --- | --- | --- | --- | --- | --- |
| | -15 ℃ | | -20 ℃ | | -25 ℃ | |
| **Mineral dust tracers** | $R$ | $P$ | $R$ | $P$ | $R$ | $P$ |
| Al ($ng/m^3$) | 0.29 | 0.27 | 0.53 | 0.03 | **0.72** | <0.01 |
| Fe ($ng/m^3$) | 0.25 | 0.36 | 0.60 | 0.01 | **0.76** | <0.01 |
| Ti ($ng/m^3$) | 0.05 | 0.85 | 0.29 | 0.28 | **0.70** | <0.01 |
| Mn ($ng/m^3$) | 0.06 | 0.83 | 0.32 | 0.22 | **0.74** | <0.01 |
| **Sea spray tracers** | $R$ | $P$ | $R$ | $P$ | $R$ | $P$ |
| $Cl^-$ ($ng/m^3$) | 0.48 | 0.06 | -0.01 | 0.97 | -0.56 | 0.02 |
| $Na^+$ ($ng/m^3$) | 0.49 | 0.06 | -0.05 | 0.86 | -0.56 | 0.02 |
| **Anthropogenic aerosol tracers** | $R$ | $P$ | $R$ | $P$ | $R$ | $P$ |
| Zn ($ng/m^3$) | 0.07 | 0.81 | 0.09 | 0.75 | 0.18 | 0.51 |
| Pb ($ng/m^3$) | 0.09 | 0.74 | 0.24 | 0.37 | 0.30 | 0.26 |
| $NH_4^+$ ($ng/m^3$) | -0.02 | 0.93 | 0.01 | 0.96 | 0.42 | 0.10 |
| nss-$SO_4^{2-}$ ($ng/m^3$) | -0.19 | 0.47 | -0.08 | 0.77 | 0.38 | 0.15 |
| $NO_3^-$ ($ng/m^3$) | 0.24 | 0.37 | 0.02 | 0.95 | 0.22 | 0.41 |





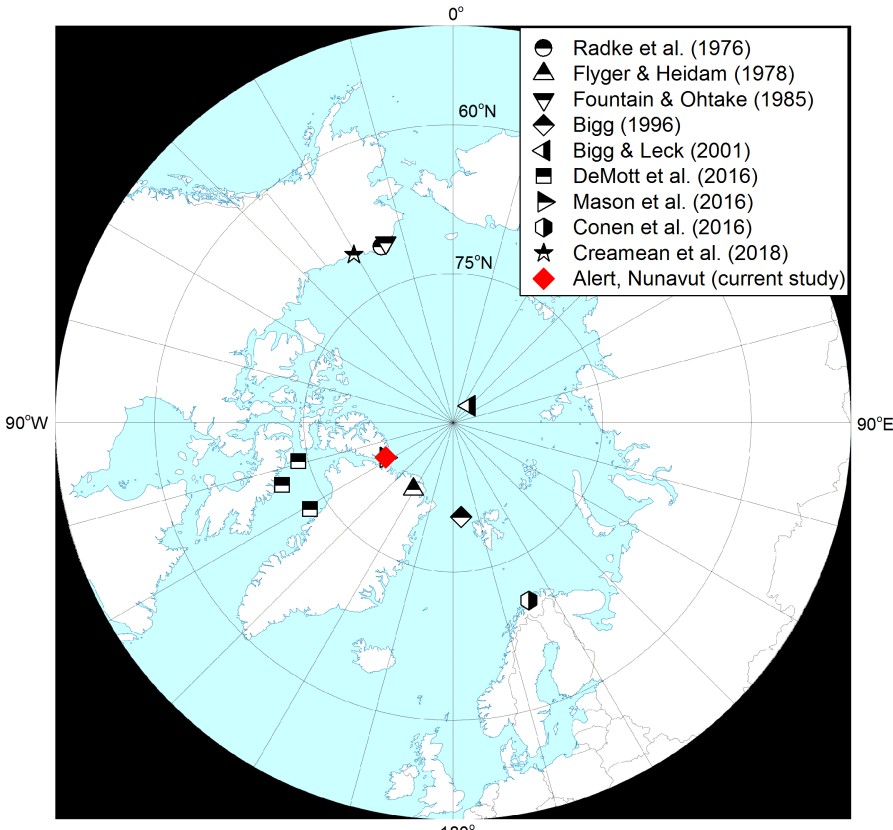

**Figure 1. The locations of previous ground-based and ship-based INP studies in the Arctic. For Bigg (1996) and Bigg & Leck (2001), samples were collected along the ship track, but only one location for each study is shown in this map. The red diamond represents the location of current study.**



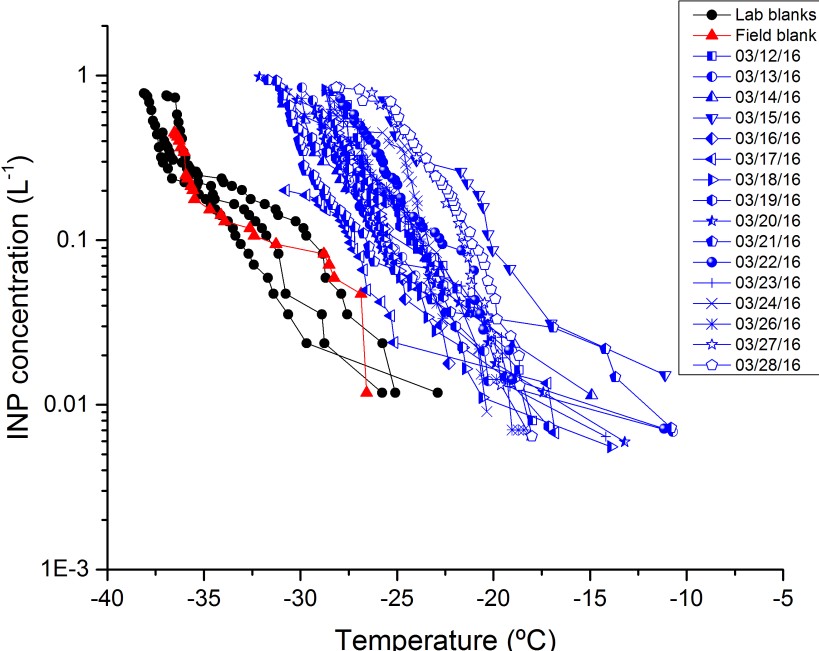

**Figure 2. The INP concentrations from each sample plotted as a function of temperature. Black represents lab blanks. Red represents the field blank, which was collected by putting the glass slides briefly in the inertia impactor without turning on the pump. Blue represents samples, and the legend gives the collection date of each sample.**



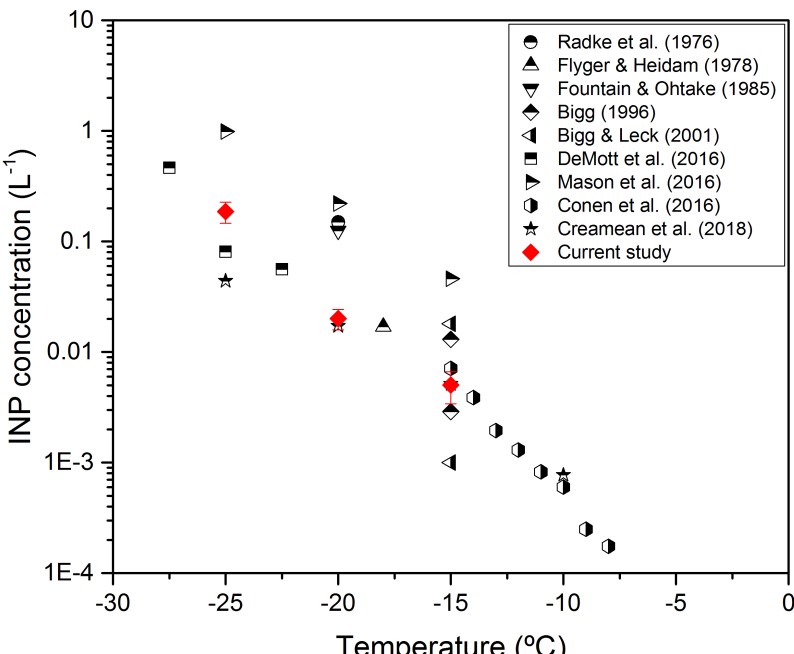

**Figure 3. INP concentrations measured at ground level or sea level in the Arctic. Black represents previous ground-based and ship-based studies, and red represents the current study. The error bars on current study data points represent the standard error of the mean.**



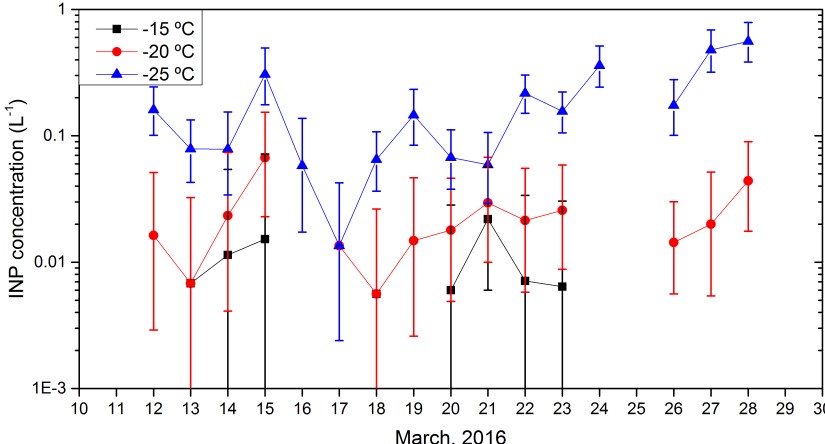

**Figure 4. The time series of INP concentrations at -15 ℃, -20 ℃, and -25 ℃. INP samples were not collected on March 25 due to time constraint. On several days, INP concentrations are shown at -25 ℃, but not at -15 ℃ and some cases not at -20 ℃. In these cases, INP concentration was below the detection limit, and hence not shown in this figure. The uncertainties in INP concentrations were calculated as described in Sect. 2.2.2.**





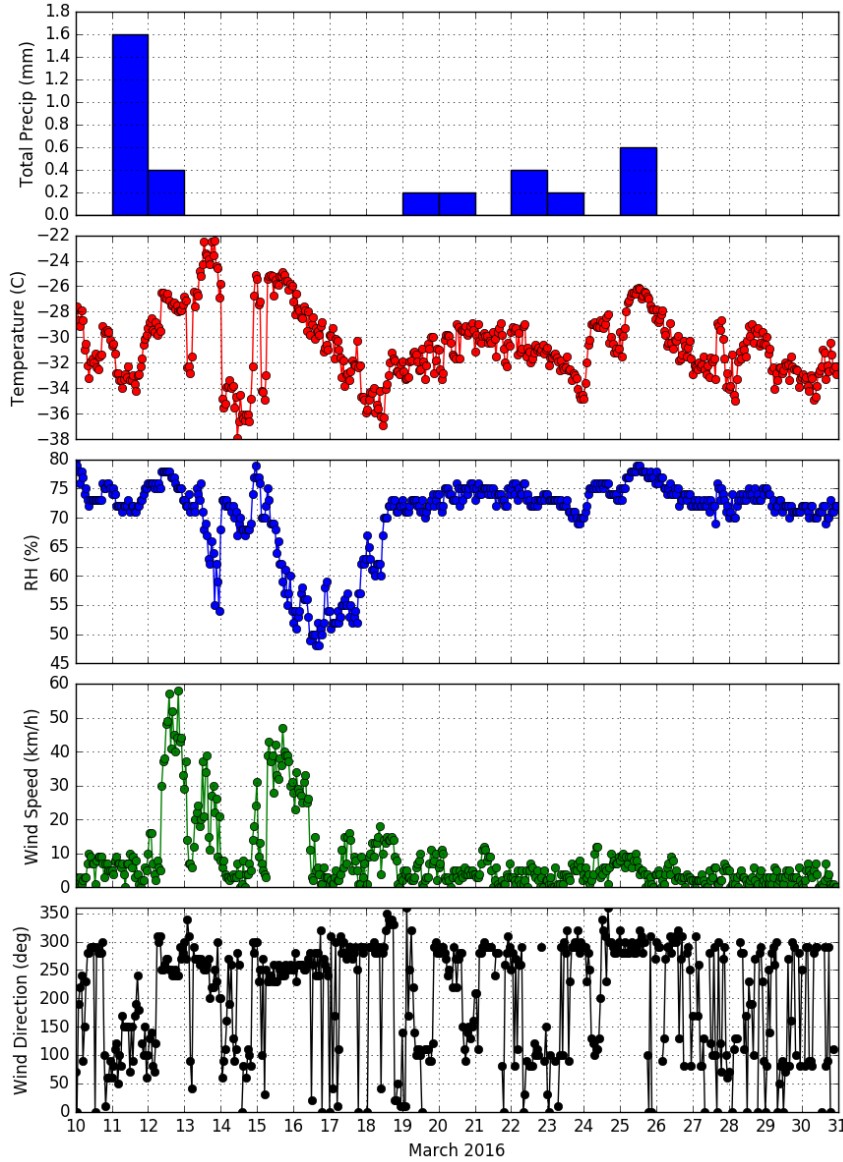

**Figure 5. The time series of meteorological parameters, including total precipitation, ambient temperature, ambient relative humidity (RH), wind speed, and wind direction. The meteorological data were retrieved from climate.weather.gc.ca. For wind direction, a value of zero denotes a calm wind, and a value of 360 denotes a wind blowing from the geographic North Pole.**




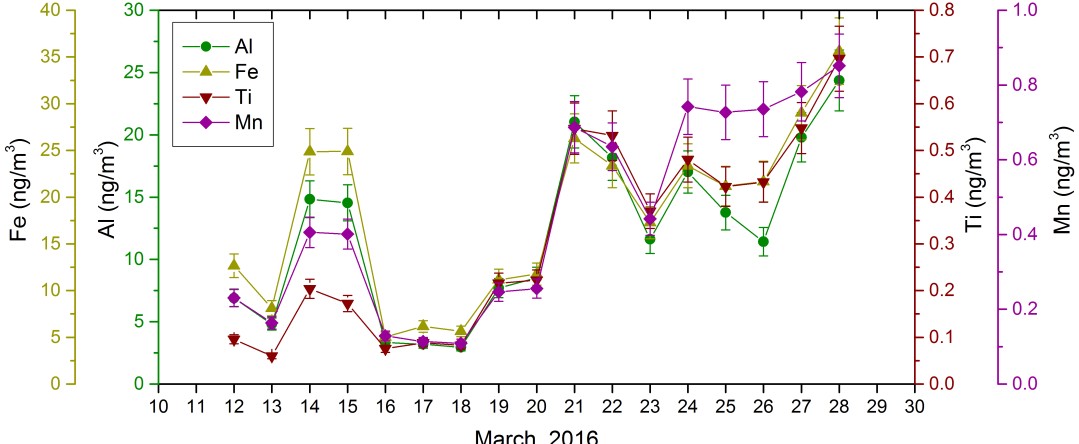

**Figure 6. The time series of mineral dust tracers (Al, Fe, Ti, and Mn).**



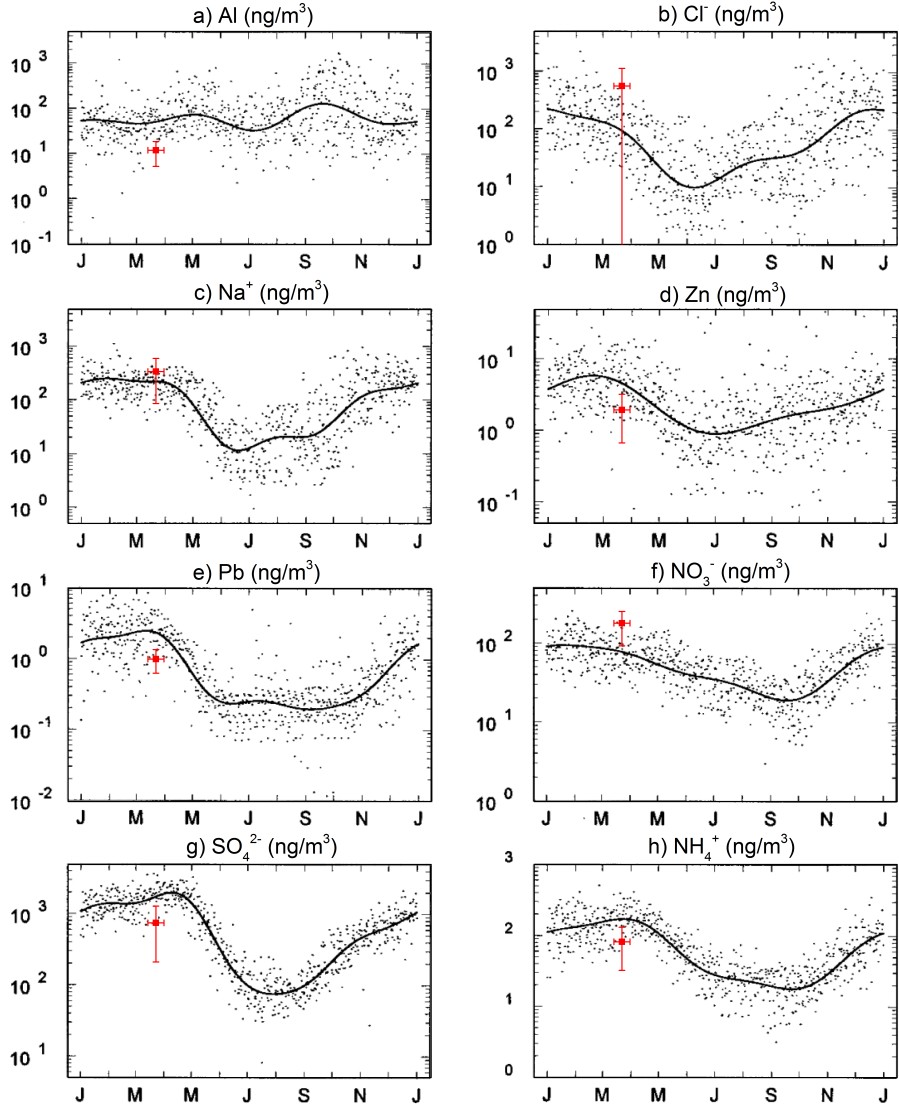

**Figure 7. Comparison between the mean mass concentrations of each of the aerosol tracers measured at the Alert site during the current study and the concentrations reported from a 15-year (1980-1995) study at the same site (Sirois and Barrie, 1999). The x-axis represents the months. The black dots represent the weekly concentrations measured by Sirois and Barrie (1999), and the blank line represents the estimated seasonal variations. The red symbols represent the mean mass concentrations measured in this study with standard deviation as y-error bars. The x-error bars represent the time period of this study. The mass concentrations of Fe, Ti and Mn were not available from the 15-year study.**



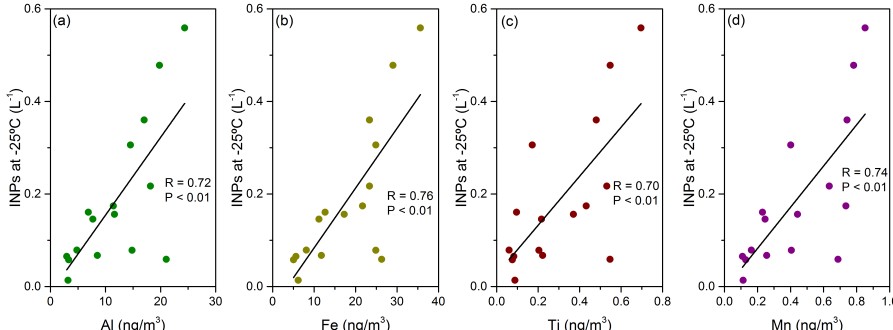

**Figure 8. Correlation plots between INP concentrations at -25 ºC and mineral dust tracers (Al, Fe, Ti, and Mn). Included in each panel are the correlation coefficient (*R*) and the probability value (*P*).**





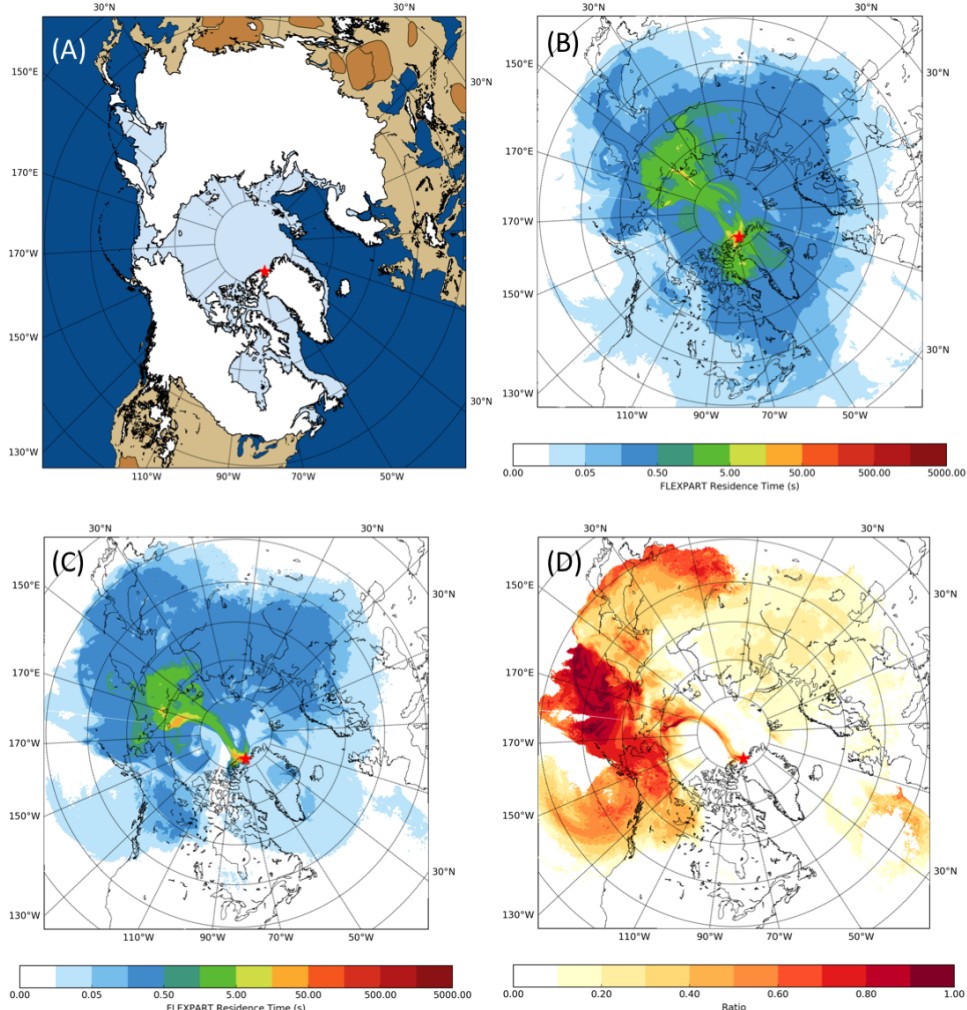

Figure 9. Panel (A) shows the surface coverage types on the first day of sampling (March 12, 2016). White represents snow, light blue represents ice, dark blue represents ocean, light brown represents land, and dark brown represents desert (data from National Snow & Ice Data Center, https://doi.org/10.7265/N52R3PMC); panel (B) is the average footprint PES plot from a 20 day FLEXPART analysis for all mineral dust sampling periods; panel (C) is the average footprint PES plot for the four highest mineral dust sampling periods, which were collected on March 21, 22, 27, and 28; panel (D) is the ratio of panel (C) to panel (B). Red star indicates the sampling location.



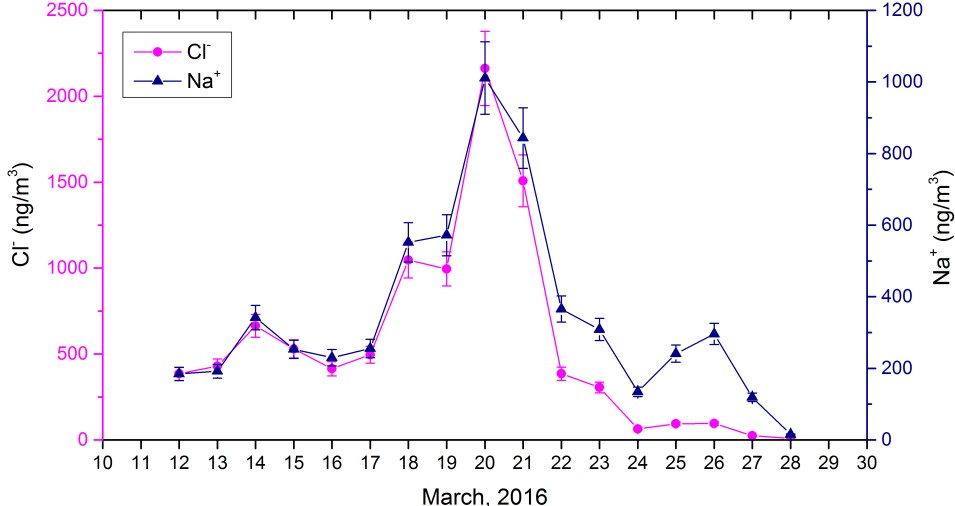

**Figure 10. The time series of sea spray tracers (Cl⁻ and Na⁺).**



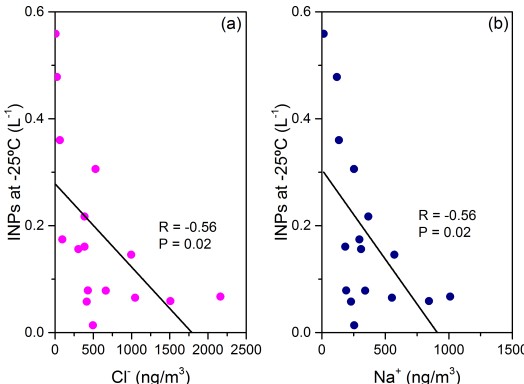

**Figure 11. Correlation plots between INP concentrations at -25 ºC and sea spray tracers (Cl⁻ and Na⁺). Included in each panel are the correlation coefficient (*R*) and the probability value (*P*).**



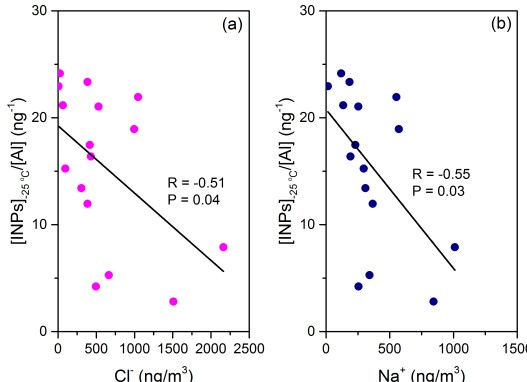

**Figure 12. Correlation plots between the ratio of INP number concentration at -25 ºC to aluminum mass concentration and sea spray tracers (Cl⁻ and Na⁺). Included in each panel are the correlation coefficient (*R*) and the probability value (*P*).**



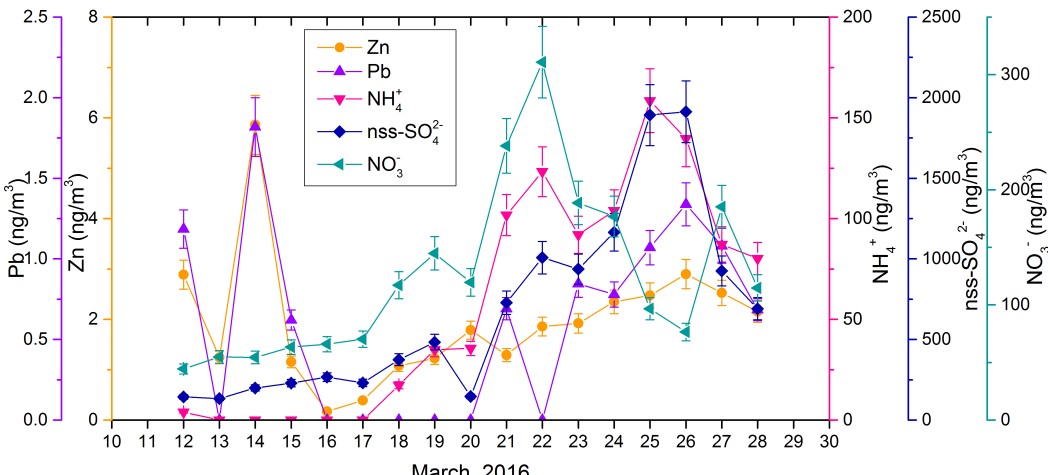

Figure 13. The time series of anthropogenic aerosol tracers (Zn, Pb, $NH_4^+$, nss-$SO_4^{2-}$, and $NO_3^-$).



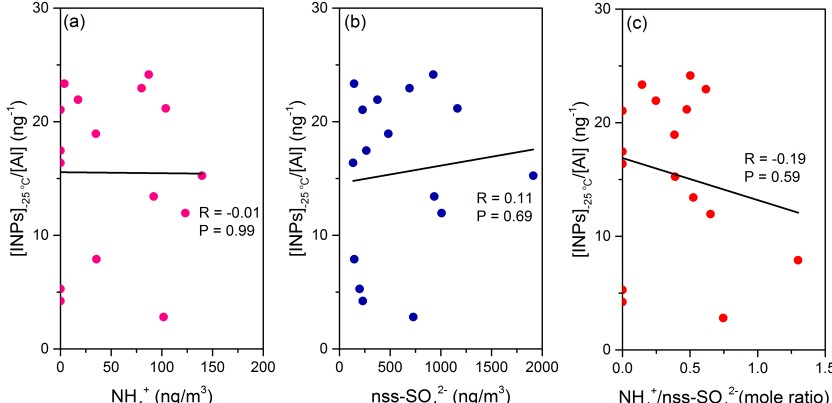

**Figure 14.** Correlation plots between the ratio of INP number concentration at -25 °C to aluminum mass concentration and anthropogenic pollution tracers ($NH_4^+$ and $nss\text{-}SO_4^{2-}$) and $NH_4^+/nss\text{-}SO_4^{2-}$ mole ratio. Included in each panel are the correlation coefficient ($R$) and the probability value ($P$).