# Peer review of "Concentrations, composition, and sources of ice-nucleating particles in the Canadian High Arctic during spring 2016"

_Atmospheric Chemistry and Physics, 2018_

## Referee Comment (RC1) · Anonymous Referee #2 · 29 Oct 2018

Review for "Concentrations, composition, and sources of ice-nucleating particles in the Canadian High Arctic during spring 2016" by Meng Si and co-authors, submitted to ACPD:

The manuscript reviewed herein is a well written, up-to-date and informative piece of work. Impactor samples were taken in the Arctic and analyzed wrt. atmospheric concentrations of ice nucleating particles. The results were brought into context with information on the chemical composition and on air mass origin. INP that were ice active at -25°C were shown to correlate with mineral dust particles, with the Gobi desert being a possible source. For INP ice active at higher temperatures, no correlation was

found, although sea spray and anthropogenic tracers were tested.

There is a number of (mostly smaller) issues listed below. Other than that, the manuscript well merits publication in ACP once these issues have been tackled.

Abstract: p1, line 17: Add "at that temperature" behind "INP population", as this is likely not the case at higher temperatures.

Also, mention that you examine immersion freezing – best do that already in the abstract, and/or maybe again later.

And either here or maybe in the methods section, mention how many separate samples you analyzed for INP concentrations. I did not know what to expect until I saw Fig. 2.

Introduction: p2, line 31 ff: It should become clear that different ice active substances are ice active at different temperatures. Sea spray, anthropogenic aerosol and mineral dust all are a bit intermingled, here. Please make this clearer.

Methods: p3, line 17: It is not clear to me why there are three circular glass slides on the second stage? It sounds as if they are there simultaneously? Or did you only use three during the whole time? Please clarify!

p4, line 20: It is crucial what happens to the RH during cooling, in terms of the effect of RH: did the droplets grow or shrink during this phase, if yes, to which extent? (They might get in touch with each other, or they might dry out, as the two extreme ends of what could happen.) Is that prevented somehow? How? A bit of text discussing this should be added here.

p4, line 28 ff: Do really several separate droplets form on one spot? How large is such a spot? You may want to give this number here.

p5, line 5 ff: Freezing of "non-spot droplets" may come from impurities on or in the glass, particularly as particles as small as 180 nm may not be very ice active. And if this is the case, your way of analyzing the data leads to an overestimation. Did you

try the experiment on a "field blank" glass (glass slides treated as the ones used for sampling, including to put them into the sampler, just without turning on the air flow)? Are those "blanks" you mention on p5, line 20 such "field blanks"? How do the data from your "blanks" compare to the number of "non-spot droplets"? And should not the counts obtained from the "blanks" be subtracted from the counts of the samples, instead of adding "non-spot droplets" to the counts of the samples?

p5, line 14 ff: The number 300 confused me. Is that the number of impactor spots that were sampled on that stage? If yes, please clarify! Also, above you discuss a number of sources for uncertainty (several droplet on one spot, merging droplets, . . .), and here now you say that the uncertainties were derived based on Koop et al. (1997). So were these issues (several droplet on one spot, merging droplets, . . .) not considered for the uncertainty?

p5, line 19: More information on the background measurements is needed. Right now, you basically only show this background data in Fig. 2, but it is not said if it had been subtracted, and depending on if you did or did not subtract it, why it was done how it was done. This should be added here.

p6, line 13 ff: I now know that it is discussed later that the daily filter samples described here have a different time resolution than the samples taken for INP analysis (p-8, line 31). This confused me when reading it for the first time, and it would be good to mention already here, that this is the case and that it is discussed later. A the later discussion of this issue, it would be good to mention if air mass trajectories were at least similar throughout the whole days when you sampled. For days when that was not the case, less correlation between INP and any information on chemical composition should be expected.

Results and discussion: p9, line 27: "The North Pacific Ocean is not likely the source of mineral dust either." The high signal in Panel D of Fig. 9 confused me in the beginning. But after looking at the plots a bit, I now assume that it originates in the fact that this

Pacific region gets its high signals as it has small signals in both, plot B and C. So it would be the result of bad statistics (dividing a small signal by yet smaller signals).

But still, would not all regions in B have to have higher concentrations as those in C? C, as I understand it, is a subset of B, and yet, there are regions in C that indicate larger concentrations, which cause the high values in the Pacific in D. - How can this be? Please explain the latter, and also mention that high values in the Pacific in "D" come from the division of small signals by small signals, if that is the case.

p10, line 12-16: I agree that you show that mineral dust contributes INP that are ice active at -25°C. You also show that INP that are ice active at higher temperatures have other sources (as they do not correlate with mineral dust). So the statement you make here is a bit twisted, as you do not show what contributed the INP that are ice active at the higher temperatures, which is the temperature range for which INP from sea spray (generally assumed to be biogenic in origin) may be ice active. And for mixed phase clouds INP active at higher temperatures might be even more important than INP active at -25°C. Please correct your statement here accordingly, particularly the last sentence.

p10, line 21 ff: It is VERY well known that salt in general causes a freezing point depression (that's why salt is put on icy roads in winter time). That is why I was somewhat surprised by that part in general, as I would not consider this a new result. There is also other older work on that, not for NaCl but for salts and other soluble material in general, on the temperature depression of ice nucleation for minerals (Koop & Zobrist, 2009, Wex et al., 2014), in which even more general ways of describing this suppression are given. Although this originates from laboratory studies, it may merit including that and mentioning that your results is not such an astonishing finding. Also, as you know the salt concentrations on the filters, it would be interesting to see how much freezing point depression could be expected from that. This is easy to do, and, albeit only being an estimate, it could show if an effect may be expected at all.

[Figure]

p11, line 17: Do you mean "at any of the three temperatures"?

p11, line 19ff: As you discuss other non-Arctic studies here, you may want to include a recent study on INP in Beijing, where it is also said that: "All these results indicate that Beijing air pollution did not increase or decrease INP concentrations in the examined temperature range (down to -25°C)." (Chen et al., 2018).

Figures Caption of Fig. 9: What do you mean by "for all mineral dust sampling periods"? The time when samples were analyzed wrt. INP concentrations? Please clarify, here or in the respective text.

Supplement There is a supplement with nothing else than one table that is not even large. This should be included in the main manuscript. It makes no sense to have a separate document for so little of information.

Literature Chen, J., Z. Wu, S. Augustin-Bauditz, S. Grawe, M. Hartmann, X. Pei, Z. Liu, D. Ji, and H. Wex (2018), Ice nucleating particle concentrations unaffected by urban air pollution in Beijing, China, Atmos. Chem. Phys., 18, 3523–3539, doi:10.5194/acp-18-3523-2018.

Koop, T., and B. Zobrist (2009), Parameterizations for ice nucleation in biological and atmospheric systems, Phys. Chem. Chem. Phys., 11(46), 10839-10850, doi:10.1039/b914289d.

Wex, H., P. J. DeMott, Y. Tobo, S. Hartmann, M. Rösch, T. Clauss, L. Tomsche, D. Niedermeier, and F. Stratmann (2014), Kaolinite particles as ice nuclei: learning from the use of different kaolinite samples and different coatings, Atmos. Chem. Phys., 14, 5529-5546, doi:10.5194/acp-14-5529-2014.

---

## Referee Comment (RC2) · Anonymous Referee #1 · 3 Nov 2018

Si et al. investigated the concentrations of ice nucleating particles (INP) at Alert, Canada for three weeks in 2016. This study measured INP concentrations in the immersion freezing using the droplet freezing technique. The presented INP concentrations are within the ranges of previous INP measurements in the Arctic. Complementary measurements of tracers for mineral dust, sea spray aerosol, and anthropogenic aerosols were also conducted. The correlation of INPs with these tracers are also investigated. The subject of this manuscript is within the scope of this journal. There are some minor issues and comments that the authors may want to address or consider in

the revision.

1. P4, L17-18, the particles were exposed to RH above water saturation, to what extend? Is there an estimation?

2. P4, L28 to P5, L11, it is not very clear how the upper and lower limits were calculated and how the freezing of non-spot droplets was considered in the INP calculation. It may be easy for reader to understand or to use such method if example can be provided in the supplement.

3. The spots of particles may contain more than one INPs. When cooling the droplet, the first or most efficient INP will trigger the freezing. Once the droplet freezes, it won't freeze again at lower temperature. This could lead to the underestimation of INP concentration. Please comment on this issue.

4. P6, L30-31, although the equation 3 is the standard method to calculate nss-so4, how the Cl depletion in sea salt particles will affect these calculations.

5. P8, L31 to P9, L2, the manuscript states this assumption, it would be useful if the authors can add a short discussion on the potential bias this assumption may lead to the correlation analysis.

6. How the non-INP affect the freezing data when dissolve in the droplets, this is due to sampling method (collecting a spot of impacted particle vs. individual particles), such as salts including NaCl? This may need additional discussion.
* * *

---

## Author Comment (AC1) · 7 Jan 2019

Prof. Barbara Ervens

Co-Editor of Atmospheric Chemistry and Physics

Dear Barbara,

Listed below are our responses to the comments from the referees of our manuscript. For clarity and visual distinction, the referee comments or questions are listed here in black and are preceded by bracketed, italicized numbers (e.g. *[1]*). Author's responses are offset in blue below each referee statement with matching numbers (e.g. *[A1]*). Page and line numbers refer to the online ACPD version. We thank the referees for carefully reading our manuscript and for their helpful comments!

Sincerely,

Allan Bertram,
Professor, Department of Chemistry
University of British Columbia

**Anonymous Referee #1**

Si et al. investigated the concentrations of ice nucleating particles (INP) at Alert, Canada for three weeks in 2016. This study measured INP concentrations in the immersion freezing using the droplet freezing technique. The presented INP concentrations are within the ranges of previous INP measurements in the Arctic. Complementary measurements of tracers for mineral dust, sea spray aerosol, and anthropogenic aerosols were also conducted. The correlations of INPs with these tracers are also investigated. The subject of this manuscript is within the scope of this journal. There are some minor issues and comments that the authors may want to address or consider in the revision.

*[1]* P4, L17-18, the particles were exposed to RH above water saturation, to what extend? Is there an estimation?

> *[A1]* A humidified gas flow with a dew point of approximately 3 ºC was passed through the flow cell. When the flow cell was at 0 ºC, the particles were exposed to a RH with respect to liquid water of approximately 115 %. To address the referee's comments, this information will be added to Sect. 2.2.2 in the revised manuscript.

*[2]* P4, L28 to P5, L11, it is not very clear how the upper and lower limits were calculated and how the freezing of non-spot droplets was considered in the INP calculation. It may be easy for reader to understand or to use such method if example can be provided in the supplement.

> *[A2]* To address the referee's comment, the description of the calculation of INP concentrations and their uncertainties will be improved in the revised manuscript.

*[3]* The spots of particles may contain more than one INPs. When cooling the droplet, the first or most efficient INP will trigger the freezing. Once the droplet freezes, it won't freeze again at lower temperature. This could lead to the underestimation of INP concentration. Please comment on this issue.

> *[A3]* Equation (1) does not underestimate the INP concentrations due to the issue raised by the referee. Equation (1) represents the cumulative nucleus spectrum or the concentrations of INPs active at all temperatures warmer than temperature T. This equation was derived by integrating the differential nucleus spectrum. The use of Eq. (1) to quantify the concentrations of INPs active at all temperature warmer than temperature T from droplet freezing experiments has previously been justified using Poisson's law and Monte Carlo simulations. To address the referee's comments, this information will be added to the revised manuscript.

*[4]* P6, L30-31, although the equation 3 is the standard method to calculate nss-so4, how the Cl depletion in sea salt particles will affect these calculations.

> *[A4]* To address the referee's comment, we have calculated [nss-SO$_4^{2-}$] using a method that does not rely on [Cl$^-$]. Specifically, we calculated [nss-SO$_4^{2-}$] using the following equation:

$$[nss - SO_4^{2-}] = [SO_4^{2-}] - 0.25[Na^+],$$

where $Na^+$ comes solely from sea salt and is not affected by Cl depletion (Balasubramanian, 2003). The difference between the concentrations of nss-$SO_4^{2-}$ based on $Cl^-$ and the concentrations based on $Na^+$ is less than 5 %. This difference will be mentioned in the revised manuscript.

*[5]* P8, L31 to P9, L2, the manuscript states this assumption, it would be useful if the authors can add a short discussion on the potential bias this assumption may lead to the correlation analysis.

*[A5]* In the revised manuscript, we will comment on the potential implications of this assumption.

*[6]* How the non-INP affect the freezing data when dissolve in the droplets, this is due to sampling method (collecting a spot of impacted particle vs. individual particles), such as salts including NaCl? This may need additional discussion.

*[A6]* The referee is correct that the method of collection and formation of droplets results in the soluble material, such as salts including NaCl, being mixed with mineral dust particles within the same droplet. In the atmosphere, however, the soluble material may not be mixed with the insoluble material. Studies of the mixing of soluble material with mineral dust during the same time of the year and at the same location are needed. To address the referee's comments, this information will be added to Sect. 3.4 in the revised manuscript.

**Anonymous Referee #2**
Review for "Concentrations, composition, and sources of ice-nucleating particles in the Canadian High Arctic during spring 2016" by Meng Si and co-authors, submitted to ACPD:

The manuscript reviewed herein is a well written, up-to-date and informative piece of work. Impactor samples were taken in the Arctic and analyzed wrt. atmospheric concentrations of ice nucleating particles. The results were brought into context with information on the chemical composition and on air mass origin. INP that were ice active at -25$^{\circ}$C were shown to correlate with mineral dust particles, with the Gobi desert being a possible source. For INP ice active at higher temperatures, no correlation was found, although sea spray and anthropogenic tracers were tested.

There is a number of (mostly smaller) issues listed below. Other than that, the manuscript well merits publication in ACP once these issues have been tackled.

*[7]* Abstract: p1, line 17: Add "at that temperature" behind "INP population", as this is likely not the case at higher temperatures.

> *[A7]* To address the referee's comment, we will specify the temperature after "INP population".

*[8]* Also, mention that you examine immersion freezing – best do that already in the abstract, and/or maybe again later.

> *[A8]* To address the referee's comment, we will indicate earlier in the abstract that we examined immersion freezing.

*[9]* And either here or maybe in the methods section, mention how many separate samples you analyzed for INP concentrations. I did not know what to expect until I saw Fig. 2.

> *[A9]* To address the referee's comment, the number of total samples collected for INP analysis will be mentioned in the Abstract and again in the Methods.

*[10]* Introduction: p2, line 31 ff: It should become clear that different ice active substances are ice active at different temperatures. Sea spray, anthropogenic aerosol and mineral dust all are a bit intermingled, here. Please make this clearer.

> *[A10]* To address the referee's comment, the introduction will be modified to make it clear that different ice active substances are ice active at different temperatures.

*[11]* Methods: p3, line 17: It is not clear to me why there are three circular glass slides on the second stage? It sounds as if they are there simultaneously? Or did you only use three during the whole time? Please clarify!

> *[A11]* Three hydrophobic glass slides were located on the second stage simultaneously. As a result, each sample consisted of three hydrophobic glass slides.

*[12]* p4, line 20: It is crucial what happens to the RH during cooling, in terms of the effect of RH: did the droplets grow or shrink during this phase, if yes, to which extent? (They might get in touch with each other, or they might dry out, as the two extreme ends of what could happen.) Is that prevented somehow? How? A bit of text discussing this should be added here.

*[A12]* If no gas is passed through the flow cell during cooling, the droplets tend to grow by condensation. To prevent the growth, a small flow of dry helium (~ 0.2 L min$^{-1}$) was passed through the flow cell during cooling. To address the referee's comments, this information will be added to the Sect. 2.2.2 in the revised manuscript.

*[13]* p4, line 28 ff: Do really several separate droplets form on one spot? How large is such a spot? You may want to give this number here.

*[A13]* The diameter of each spot was approximately 110 μm. This information will be added to Sect. 2.2.2 in the revised manuscript.

*[14]* p5, line 5 ff: Freezing of "non-spot droplets" may come from impurities on or in the glass, particularly as particles as small as 180 nm may not be very ice active. And if this is the case, your way of analyzing the data leads to an overestimation. Did you try the experiment on a "field blank" glass (glass slides treated as the ones used for sampling, including to put them into the sampler, just without turning on the air flow)? Are those "blanks" you mention on p5, line 20 such "field blanks"? How do the data from your "blanks" compare to the number of "non-spot droplets"? And should not the counts obtained from the "blanks" be subtracted from the counts of the samples, instead of adding "non-spot droplets" to the counts of the samples?

*[A14]* The blanks mentioned on P5, line 20 and shown in Fig. 2 refer to field blanks (treated exactly as the referee described) and lab blanks (glass slides that were cleaned in the lab the same way as the ones used for sampling in the field). Lab and field blanks shown in Fig. 2 are consistent with lab and field blanks determined in another study using exactly the same techniques described in the current study (Irish et al., 2018). In short, rarely do we see freezing at -25 ºC or above in the field or lab blanks. In the current study, we report INP concentrations at ≥ -25 ºC. Thus, any freezing events of non-spot droplets that happened at ≥ -25 ºC were most likely not from impurities on or in the glass substrates. We assume that non-spot droplet freezing is due to INPs < 0.18 μm not being focused into the spots, or that a small fraction of the INPs ≥ 0.18 μm may not be concentrated into the spots due to rebound from the substrate. To address the referee's comments, this information will be added to the revised manuscript.

*[15]* p5, line 14 ff: The number 300 confused me. Is that the number of impactor spots that were sampled on that stage? If yes, please clarify! Also, above you discuss a number of sources for uncertainty (several droplet on one spot, merging droplets, . . .), and here now you say that the uncertainties were derived based on Koop et al. (1997). So were

these issues (several droplet on one spot, merging droplets, . . .) not considered for the uncertainty?

> *[A15]* 300 is the number of nozzles in the nozzle plate of the impactor, and consequently, the number of spots generated on the second impactor stage below the nozzle plate. The uncertainties based on Koop et al. (1997) account for the uncertainties due to the limited number of freezing events detected. The overall uncertainty reported in the manuscript includes this uncertainty as well as the uncertainties discussed earlier. This will be clarified in the revised manuscript.

*[16]* p5, line 19: More information on the background measurements is needed. Right now, you basically only show this background data in Fig. 2, but it is not said if it had been subtracted, and depending on if you did or did not subtract it, why it was done how it was done. This should be added here.

> *[A16]* To address the referee's comments, more information about the background measurements (i.e. blanks) will be added to Sect. 2.2.2 in the revised manuscript.

*[17]* p6, line 13 ff: I now know that it is discussed later that the daily filter samples described here have a different time resolution than the samples taken for INP analysis (p-8, line 31). This confused me when reading it for the first time, and it would be good to mention already here, that this is the case and that it is discussed later. A the later discussion of this issue, it would be good to mention if air mass trajectories were at least similar throughout the whole days when you sampled. For days when that was not the case, less correlation between INP and any information on chemical composition should be expected.

> *[A17]* Thank you for pointing this out. We will mention earlier (P6, Line 13) that daily filter samples had a different time resolution than the samples taken for INP analysis. In addition, we will add to the Supplement air mass trajectories initiated every 2 h to illustrate the variability in air mass trajectories over the 24 h filter sampling periods.

*[18]* Results and discussion: p9, line 27: "The North Pacific Ocean is not likely the source of mineral dust either." The high signal in Panel D of Fig. 9 confused me in the beginning. But after looking at the plots a bit, I now assume that it originates in the fact that this Pacific region gets its high signals as it has small signals in both, plot B and C. So it would be the result of bad statistics (dividing a small signal by yet smaller signals).

But still, would not all regions in B have to have higher concentrations as those in C? C, as I understand it, is a subset of B, and yet, there are regions in C that indicate larger concentrations, which cause the high values in the Pacific in D. - How can this be? Please explain the latter, and also mention that high values in the Pacific in "D" come from the division of small signals by small signals, if that is the case.

> *[A18]* This section will be revised for clarity.

*[19]* p10, line 12-16: I agree that you show that mineral dust contributes INP that are ice

active at -25°C. You also show that INP that are ice active at higher temperatures have other sources (as they do not correlate with mineral dust). So the statement you make here is a bit twisted, as you do not show what contributed the INP that are ice active at the higher temperatures, which is the temperature range for which INP from sea spray (generally assumed to be biogenic in origin) may be ice active. And for mixed phase clouds INP active at higher temperatures might be even more important than INP active at -25°C. Please correct your statement here accordingly, particularly the last sentence.

*[A19]* To address the referee's comments, we will specify the temperature to make the statement more accurate.

*[20]* p10, line 21 ff: It is VERY well known that salt in general causes a freezing point depression (that's why salt is put on icy roads in winter time). That is why I was somewhat surprised by that part in general, as I would not consider this a new result. There is also other older work on that, not for NaCl but for salts and other soluble material in general, on the temperature depression of ice nucleation for minerals (Koop & Zobrist, 2009, Wex et al., 2014), in which even more general ways of describing this suppression are given. Although this originates from laboratory studies, it may merit including that and mentioning that your results is not such an astonishing finding. Also, as you know the salt concentrations on the filters, it would be interesting to see how much freezing point depression could be expected from that. This is easy to do, and, albeit only being an estimate, it could show if an effect may be expected at all.

*[A20]* To address the referee's comments, we have estimated how much the NaCl can decrease the freezing temperatures of mineral dust by the well-known freezing point depression mechanism, which involves the salt changing the water activity in solution. The maximum concentration of NaCl in our freezing experiments was ~ 0.03 mol L$^{-1}$, which would only cause a freezing point depression of ~ 0.1 ºC. Since this number is within the uncertainty of our measured freezing temperatures, it is too small to explain the negative correlations shown in Fig. 11 and Fig. 12. On the other hand, very recent studies have illustrated that trace amounts of NaCl can lower the freezing temperature of feldspar (a type of mineral dust) by surface specific interactions between the solute (NaCl) and the mineral dust surface, and the decrease in freezing temperature by this mechanism is more than expected based on traditional freezing point depression mechanism (Whale et al., 2018). Additional discussion will be added to Sect. 3.4 in the revised manuscript to make it clear that our results cannot be explained by the traditional freezing point depression mechanism.

*[21]* p11, line 17: Do you mean "at any of the three temperatures"?

*[A21]* This statement will be revised to improve clarity.

*[22]* p11, line 19ff: As you discuss other non-Arctic studies here, you may want to include a recent study on INP in Beijing, where it is also said that: "All these results indicate that Beijing air pollution did not increase or decrease INP concentrations in the examined temperature range (down to -25°C)." (Chen et al., 2018).

*[A22]* Thank you for the suggestion. This reference will be added to the revised manuscript.

*[23]* Figures Caption of Fig. 9: What do you mean by "for all mineral dust sampling periods"? The time when samples were analyzed wrt. INP concentrations? Please clarify, here or in the respective text.

*[A23]* This figure caption will be revised to improve clarity.

*[24]* Supplement There is a supplement with nothing else than one table that is not even large. This should be included in the main manuscript. It makes no sense to have a separate document for so little of information.

*[A24]* Additional plots will be added to the Supplement to address Comment *[17]* above. Hence, a Supplement will be necessary.

Literature Chen, J., Z. Wu, S. Augustin-Bauditz, S. Grawe, M. Hartmann, X. Pei, Z. Liu, D. Ji, and H. Wex (2018), Ice nucleating particle concentrations unaffected by urban air pollution in Beijing, China, Atmos. Chem. Phys., 18, 3523–3539, doi:10.5194/acp-18-3523-2018.

Koop, T., and B. Zobrist (2009), Parameterizations for ice nucleation in biological and atmospheric systems, Phys. Chem. Chem. Phys., 11(46), 10839-10850, doi:10.1039/b914289d.

Wex, H., P. J. DeMott, Y. Tobo, S. Hartmann, M. Rösch, T. Clauss, L. Tomsche, D. Niedermeier, and F. Stratmann (2014), Kaolinite particles as ice nuclei: learning from the use of different kaolinite samples and different coatings, Atmos. Chem. Phys., 14, 5529-5546, doi:10.5194/acp-14-5529-2014.

**References:**

Balasubramanian, R.: Comprehensive characterization of PM 2.5 aerosols in Singapore, J. Geophys. Res., 108(D16), 4523, doi:10.1029/2002JD002517, 2003.

Irish, V. E., Hanna, S. J., Willis, M. D., China, S., Thomas, J. L., Wentzell, J. J. B., Cirisan, A., Si, M., Leaitch, W. R., Murphy, J. G., Abbatt, J. P. D., Laskin, A., Girard, E. and Bertram, A. K.: Ice nucleating particles in the marine boundary layer in the Canadian Arctic during summer 2014, Atmos. Chem. Phys. Discuss., 1–25, doi:10.5194/acp-2018-735, 2018.

Whale, T. F., Holden, M. A., Wilson, T. W., O'Sullivan, D. and Murray, B. J.: The enhancement and suppression of immersion mode heterogeneous ice-nucleation by solutes, Chem. Sci., 9(17), 4142–4151, doi:10.1039/C7SC05421A, 2018.

---

## Author Response (AR1)

Prof. Barbara Ervens

Co-Editor of Atmospheric Chemistry and Physics

Dear Barbara,

Listed below are our responses to the comments from the referees of our manuscript. For clarity and visual distinction, the referee comments or questions are listed here in black and are preceded by bracketed, italicized numbers (e.g. *[1]*). Author's responses are offset in blue below each referee statement with matching numbers (e.g. *[A1]*). Page and line numbers refer to the online ACPD version. We thank the referees for carefully reading our manuscript and for their helpful comments!

Sincerely,

Allan Bertram,
Professor, Department of Chemistry
University of British Columbia

**Anonymous Referee #1**

Si et al. investigated the concentrations of ice nucleating particles (INP) at Alert, Canada for three weeks in 2016. This study measured INP concentrations in the immersion freezing using the droplet freezing technique. The presented INP concentrations are within the ranges of previous INP measurements in the Arctic. Complementary measurements of tracers for mineral dust, sea spray aerosol, and anthropogenic aerosols were also conducted. The correlations of INPs with these tracers are also investigated. The subject of this manuscript is within the scope of this journal. There are some minor issues and comments that the authors may want to address or consider in the revision.

*[1]* P4, L17-18, the particles were exposed to RH above water saturation, to what extend? Is there an estimation?

> *[A1]* A humidified gas flow with a dew point of approximately 3 ºC was passed through the flow cell. When the flow cell was at 0 ºC, the particles were exposed to a RH with respect to liquid water of approximately 115 %. To address the referee's comments, this information has been added to Sect. 2.2.2 in the revised manuscript.

*[2]* P4, L28 to P5, L11, it is not very clear how the upper and lower limits were calculated and how the freezing of non-spot droplets was considered in the INP calculation. It may be easy for reader to understand or to use such method if example can be provided in the supplement.

> *[A2]* To address the referee's comment, the description of the calculation of INP concentrations and their uncertainties has been improved in the revised manuscript. Specifically, the following equation has been added to illustrate how the freezing of non-spot droplets was considered in the INP calculation:
>
> $$\#INP(T) = \left(-ln\left(\frac{N_{us}(T)}{N_s}\right)\right)N_s + N_{ns}$$
>
> where $N_{ns}$ is the number of frozen non-spot droplets. Please see Sect. 2.2.2 for details.

*[3]* The spots of particles may contain more than one INPs. When cooling the droplet, the first or most efficient INP will trigger the freezing. Once the droplet freezes, it won't freeze again at lower temperature. This could lead to the underestimation of INP concentration. Please comment on this issue.

> *[A3]* Equation (1) does not underestimate the INP concentrations due to the issue raised by the referee. Equation (1) represents the cumulative nucleus spectrum or the concentrations of INPs active at all temperatures $\geq T$. The use of Eq. (1) to quantify the concentrations of INPs active at all temperatures $\geq T$ from droplet freezing experiments has been justified using Poisson's law and Monte Carlo simulations. To address the referee's comments, this information has been added to the revised manuscript.

*[4]* P6, L30-31, although the equation 3 is the standard method to calculate nss-so4, how the Cl depletion in sea salt particles will affect these calculations.

*[A4]* To address the referee's comment, we have calculated $[nss\text{-}SO_4^{2-}]$ using a method that does not rely on $[Cl^-]$. Specifically, we calculated $[nss\text{-}SO_4^{2-}]$ based on $Na^+$, where $Na^+$ comes solely from sea salt and is not affected by Cl depletion (Balasubramanian, 2003). The difference between the concentrations of $nss\text{-}SO_4^{2-}$ based on $Cl^-$ and the concentrations based on $Na^+$ is less than 5 %. This difference has been added to Sect. 2.4 in the revised manuscript.

*[5]* P8, L31 to P9, L2, the manuscript states this assumption, it would be useful if the authors can add a short discussion on the potential bias this assumption may lead to the correlation analysis.

*[A5]* In the revised manuscript, we have commented on the potential implications of this assumption. Specifically, the following statement has been added to Sect. 3.3 in the revised manuscript:

"…If this assumption is not correct, correlation coefficients between INPs and aerosol tracers will be smaller than expected. Shown in Fig. S1 are air mass back trajectories initiated every 2h during each 24h quartz filter sampling period. The back trajectories suggest that for a large fraction of the samples, the source of the air masses during INP sampling was similar to the source of the air masses measured during quartz filter sampling."

*[6]* How the non-INP affect the freezing data when dissolve in the droplets, this is due to sampling method (collecting a spot of impacted particle vs. individual particles), such as salts including NaCl? This may need additional discussion.

*[A6]* The referee is correct that the method of collection and formation of droplets results in the soluble material, such as salts including NaCl, being mixed with mineral dust particles within the same droplets. However, in the atmosphere, the soluble material, such as NaCl, may not be internally mixed with mineral dust. Studies of the mixing state of soluble material with mineral dust at Alert during the same time of the year are needed. To address the referee's comments, additional discussion has been added to Sect. 3.4 and Conclusions in the revised manuscript.

**Anonymous Referee #2**
Review for "Concentrations, composition, and sources of ice-nucleating particles in the Canadian High Arctic during spring 2016" by Meng Si and co-authors, submitted to ACPD:

The manuscript reviewed herein is a well written, up-to-date and informative piece of work. Impactor samples were taken in the Arctic and analyzed wrt. atmospheric concentrations of ice nucleating particles. The results were brought into context with information on the chemical composition and on air mass origin. INP that were ice active

at -25°C were shown to correlate with mineral dust particles, with the Gobi desert being a possible source. For INP ice active at higher temperatures, no correlation was found, although sea spray and anthropogenic tracers were tested.

There is a number of (mostly smaller) issues listed below. Other than that, the manuscript well merits publication in ACP once these issues have been tackled.

*[7]* Abstract: p1, line 17: Add "at that temperature" behind "INP population", as this is likely not the case at higher temperatures.

> *[A7]* To address the referee's comment, we have specified the temperature after "INP population".

*[8]* Also, mention that you examine immersion freezing – best do that already in the abstract, and/or maybe again later.

> *[A8]* To address the referee's comment, we have indicated earlier in the abstract that we examined immersion freezing.

*[9]* And either here or maybe in the methods section, mention how many separate samples you analyzed for INP concentrations. I did not know what to expect until I saw Fig. 2.

> *[A9]* To address the referee's comment, the total number of INP samples reported in current study has been mentioned in the Abstract and again in the Methods.

*[10]* Introduction: p2, line 31 ff: It should become clear that different ice active substances are ice active at different temperatures. Sea spray, anthropogenic aerosol and mineral dust all are a bit intermingled, here. Please make this clearer.

> *[A10]* To address the referee's comment, the introduction has been modified to make it clear that different ice active substances are ice active at different temperatures. Specifically, the sentences have been modified to the following:
>
> "These data were used to determine if mineral dust, sea spray aerosol, and anthropogenic aerosol are a major contributor to the INP population at -15, -20, and -25 °C in the Canadian High Arctic during spring. Studies as a function of temperature are necessary since different types of aerosols are ice-active at different temperatures. Although a few studies have identified mineral dust particles as an

important contributor to the INP population in the Arctic during spring, additional studies are needed to determine how often mineral dust is an important contributor. The measurements reported here together with particle dispersion modelling were also used to assess the source of the INPs at a freezing temperature of -25 ºC."

*[11]* Methods: p3, line 17: It is not clear to me why there are three circular glass slides on the second stage? It sounds as if they are there simultaneously? Or did you only use three during the whole time? Please clarify!

*[A11]* Three hydrophobic glass slides were located on the second stage simultaneously. As a result, each collected sample consisted of three hydrophobic glass slides. This has been clarified in Sect. 2.2.1 in the revised manuscript.

*[12]* p4, line 20: It is crucial what happens to the RH during cooling, in terms of the effect of RH: did the droplets grow or shrink during this phase, if yes, to which extent? (They might get in touch with each other, or they might dry out, as the two extreme ends of what could happen.) Is that prevented somehow? How? A bit of text discussing this should be added here.

*[A12]* If no gas is passed through the flow cell during cooling, the droplets tend to grow by condensation. To prevent the growth, a small flow of dry helium (~ 0.2 L min$^{-1}$) was passed through the flow cell during cooling. To address the referee's comments, this information has been added to Sect. 2.2.2 in the revised manuscript.

*[13]* p4, line 28 ff: Do really several separate droplets form on one spot? How large is such a spot? You may want to give this number here.

*[A13]* The diameter of each spot was approximately 110 μm. This information has been added to Sect. 2.2.2 in the revised manuscript.

*[14]* p5, line 5 ff: Freezing of "non-spot droplets" may come from impurities on or in the glass, particularly as particles as small as 180 nm may not be very ice active. And if this is the case, your way of analyzing the data leads to an overestimation. Did you try the experiment on a "field blank" glass (glass slides treated as the ones used for sampling, including to put them into the sampler, just without turning on the air flow)? Are those "blanks" you mention on p5, line 20 such "field blanks"? How do the data from your "blanks" compare to the number of "non-spot droplets"? And should not the counts obtained from the "blanks" be subtracted from the counts of the samples, instead of adding "non-spot droplets" to the counts of the samples?

*[A14]* The blanks mentioned on P5, line 20 and shown in Fig. 2 refer to field blanks (treated exactly as the referee described) and lab blanks (glass slides that were cleaned in the lab the same way as the ones used for sampling in the field). Lab and field blanks shown in Fig. 2 are consistent with lab and field blanks determined in another study using exactly the same techniques as described in the current study (Irish et al., 2018). In short, rarely do we see freezing at -25 ºC or above in the field or lab blanks. In the current study, we report INP concentrations at ≥ -25 ºC. Thus, any freezing events of non-spot droplets that happened at ≥ -25 ºC were most likely

not from impurities on or in the glass substrates. We assume that non-spot droplet freezing may have been due to INPs < 0.18 μm in diameter not focused into the spots or a small fraction of INPs ≥ 0.18 μm not concentrated into the spots due to rebound from the hydrophobic glass slides. To address the referee's comments, this information has been added to Sect. 2.2.2 and Sect. 3.1 in the revised manuscript.

*[15]* p5, line 14 ff: The number 300 confused me. Is that the number of impactor spots that were sampled on that stage? If yes, please clarify! Also, above you discuss a number of sources for uncertainty (several droplet on one spot, merging droplets, . . .), and here now you say that the uncertainties were derived based on Koop et al. (1997). So were these issues (several droplet on one spot, merging droplets, . . .) not considered for the uncertainty?

*[A15]* 300 is the number of nozzles in the nozzle plate of the impactor, and consequently, the number of spots generated on the second impactor stage below the nozzle plate. The uncertainties based on Koop et al. (1997) account for the uncertainty associated with the limited number of freezing events detected. The overall uncertainty in the concentrations of INPs reported in the manuscript includes this uncertainty as well as the uncertainties discussed earlier. This has been clarified in Sect. 2.2.2 in the revised manuscript.

*[16]* p5, line 19: More information on the background measurements is needed. Right now, you basically only show this background data in Fig. 2, but it is not said if it had been subtracted, and depending on if you did or did not subtract it, why it was done how it was done. This should be added here.

*[A16]* To address the referee's comments, more information about the background measurements (i.e. blanks) has been added to the revised manuscript. Specifically, the following sentences have been added to Sect. 2.2.2:

"One field blank was collected by treating the slides in the same manner as the sample hydrophobic slides including locating the hydrophobic slides in the impactor, except that the pump was not turned on. Lab blanks refer to slides cleaned in the same manner as the ones used for sampling in the field but not sent to the field."

And to Sect. 3.1:

"For the remainder of this document, we focus on INP concentrations at -15, -20, and -25 ºC. … Since freezing was rarely observed at ≥ -25 ºC in the blank experiments, the INP concentrations determined from the blank experiments were not subtracted from the INP concentrations reported in this study"

*[17]* p6, line 13 ff: I now know that it is discussed later that the daily filter samples described here have a different time resolution than the samples taken for INP analysis (p-8, line 31). This confused me when reading it for the first time, and it would be good to mention already here, that this is the case and that it is discussed later. A the later discussion of this issue, it would be good to mention if air mass trajectories were at least

similar throughout the whole days when you sampled. For days when that was not the case, less correlation between INP and any information on chemical composition should be expected.

> *[A17]* Thank you for pointing this out. The following statement has been added to Sect. 2.4 where the filter sampling time was first mentioned:
>
> "Hence, collection times for tracer measurements (~ 24h) were different than the collection times for INP measurements (~ 2h). The implications of the different collection times are discussed in Sect. 3.3."
>
> In addition, air mass trajectories initiated every 2h to illustrate the variability in air mass trajectories over the 24h quartz filter sampling periods have been added to the Supplement (Fig. S1).

*[18]* Results and discussion: p9, line 27: "The North Pacific Ocean is not likely the source of mineral dust either." The high signal in Panel D of Fig. 9 confused me in the beginning. But after looking at the plots a bit, I now assume that it originates in the fact that this Pacific region gets its high signals as it has small signals in both, plot B and C. So it would be the result of bad statistics (dividing a small signal by yet smaller signals).

But still, would not all regions in B have to have higher concentrations as those in C? C, as I understand it, is a subset of B, and yet, there are regions in C that indicate larger concentrations, which cause the high values in the Pacific in D. - How can this be? Please explain the latter, and also mention that high values in the Pacific in "D" come from the division of small signals by small signals, if that is the case.

> *[A18]* The statement that "The North Pacific Ocean is not likely the source of mineral dust" is because that the Pacific Ocean is not considered as a traditional source of mineral dust. This information has been added to the revised manuscript. Besides, the description of Fig. 9 has been revised for clarity. Specifically, the following has been added to the revised manuscript:
>
> "…Figure 9(B) shows the average footprint PES plot for all samples, and Fig. 9(C) shows the average footprint PES plot for the four samples with the highest mineral dust concentrations, which were collected on March 21, 22, 27, and 28. Note, since average footprint PES plots are shown in Fig. 9(B) and 9(C), Fig. 9(C) is not a subset of Fig. 9(B). Shown in Fig. 9(D) is a ratio plot of the sum of footprint PES plots of the four samples with the highest mineral dust concentrations to the sum of footprint PES plots of all samples. These types of ratio plots are often used as a sensitive method to identify the source regions of a component under investigation (Hirdman et al., 2010). …"

*[19]* p10, line 12-16: I agree that you show that mineral dust contributes INP that are ice active at -25°C. You also show that INP that are ice active at higher temperatures have other sources (as they do not correlate with mineral dust). So the statement you make here is a bit twisted, as you do not show what contributed the INP that are ice active at

the higher temperatures, which is the temperature range for which INP from sea spray (generally assumed to be biogenic in origin) may be ice active. And for mixed phase clouds INP active at higher temperatures might be even more important than INP active at -25°C. Please correct your statement here accordingly, particularly the last sentence.

*[A19]* To address the referee's comments, we have specified the temperature to make the statement more accurate. Specifically, the statement has been modified to:

"Our results suggest that mineral dust is a more important source of INPs at -25 ºC than sea spray aerosol for the time and location studied."

*[20]* p10, line 21 ff: It is VERY well known that salt in general causes a freezing point depression (that's why salt is put on icy roads in winter time). That is why I was somewhat surprised by that part in general, as I would not consider this a new result. There is also other older work on that, not for NaCl but for salts and other soluble material in general, on the temperature depression of ice nucleation for minerals (Koop & Zobrist, 2009, Wex et al., 2014), in which even more general ways of describing this suppression are given. Although this originates from laboratory studies, it may merit including that and mentioning that your results is not such an astonishing finding. Also, as you know the salt concentrations on the filters, it would be interesting to see how much freezing point depression could be expected from that. This is easy to do, and, albeit only being an estimate, it could show if an effect may be expected at all.

*[A20]* To address the referee's comments, we have estimated how much the NaCl can decrease the freezing temperatures of mineral dust by the traditional freezing point depression mechanism, which involves the salt changing the water activity in solution. The maximum concentration of NaCl in our freezing experiments was ~ 0.03 mol L$^{-1}$, which would only cause a freezing point depression of ~ 0.1 ºC. Since this number is within the uncertainty of our measured freezing temperatures, it is too small to explain the negative correlations shown in Fig. 11. On the other hand, very recent studies have illustrated that trace amounts of NaCl can lower the freezing temperature of feldspar (a type of mineral dust) by surface specific interactions between the solute (NaCl) and the mineral dust surface, and the decrease in freezing temperature by this mechanism is more than expected based on traditional freezing point depression mechanism (Whale et al., 2018). Additional discussion has been added to Sect. 3.4 in the revised manuscript to make it clear that our results cannot be explained by the traditional freezing point depression mechanism.

*[21]* p11, line 17: Do you mean "at any of the three temperatures"?

*[A21]* This statement has been revised to improve clarity.

*[22]* p11, line 19ff: As you discuss other non-Arctic studies here, you may want to include a recent study on INP in Beijing, where it is also said that: "All these results indicate that Beijing air pollution did not increase or decrease INP concentrations in the examined temperature range (down to -25°C)." (Chen et al., 2018).

*[A22]* Thank you for the suggestion. This reference has been added to the revised manuscript.

*[23]* Figures Caption of Fig. 9: What do you mean by "for all mineral dust sampling periods"? The time when samples were analyzed wrt. INP concentrations? Please clarify, here or in the respective text.

*[A23]* This figure caption has been revised to "panel (B) is the average footprint PES plot from a 20 day FLEXPART analysis for all samples" to improve clarity.

*[24]* Supplement There is a supplement with nothing else than one table that is not even large. This should be included in the main manuscript. It makes no sense to have a separate document for so little of information.

*[A24]* Additional plot and text have been added to the Supplement. Hence, a Supplement is necessary.

Literature Chen, J., Z. Wu, S. Augustin-Bauditz, S. Grawe, M. Hartmann, X. Pei, Z. Liu, D. Ji, and H. Wex (2018), Ice nucleating particle concentrations unaffected by urban air pollution in Beijing, China, Atmos. Chem. Phys., 18, 3523–3539, doi:10.5194/acp-18-3523-2018.

Koop, T., and B. Zobrist (2009), Parameterizations for ice nucleation in biological and atmospheric systems, Phys. Chem. Chem. Phys., 11(46), 10839-10850, doi:10.1039/b914289d.

Wex, H., P. J. DeMott, Y. Tobo, S. Hartmann, M. Rösch, T. Clauss, L. Tomsche, D. Niedermeier, and F. Stratmann (2014), Kaolinite particles as ice nuclei: learning from the use of different kaolinite samples and different coatings, Atmos. Chem. Phys., 14, 5529-5546, doi:10.5194/acp-14-5529-2014.

[Figure]

**Figure 10. The time series of sea spray tracers (Cl⁻ and Na⁺).**

[Figure]

**Figure 11. Correlation plots between INP concentrations at -25 °C and sea spray tracers (Cl⁻ and Na⁺). Included in each panel are the correlation coefficient (*R*) and the probability value (*P*).**

[Figure]

**Figure 12. Correlation plots between the ratio of INP number concentration at -25 ℃ to aluminum mass concentration and sea spray tracers (Cl⁻ and Na⁺). Included in each panel are the correlation coefficient (*R*) and the probability value (*P*).**

[Figure]

**Figure 13. The time series of anthropogenic aerosol tracers (Zn, Pb, NH$_4^+$, nss-SO$_4^{2-}$, and NO$_3^-$).**

[Figure]

**Figure 14.** Correlation plots between the ratio of INP number concentration at -25 °C to aluminum mass concentration and anthropogenic pollution tracers ($NH_4^+$ and $nss-SO_4^{2-}$) and $NH_4^+$/ $nss-SO_4^{2-}$ mole ratio. Included in each panel are the correlation coefficient (*R*) and the probability value (*P*).

**Supplemental Information**

**S1 Back trajectories**

For each high-volume (i.e. quartz filter) sampling period (~ 24h), 10-day back trajectories were calculated using the HYSPLIT4 (Hybrid Single-Particle Lagrangian Integrated Trajectory) model of the NOAA Air resources Laboratory (Stein et al., 2015). The GDAS (Global Data Assimilation System) 1° meteorological data were used as input. Back trajectories were initiated at the beginning of each quartz filter sampling period and at every 2h until the end of the sampling period. The initiation height was 10m a.g.l.. The results are shown in Fig. S1.

**Table S1. Results of linear correlation analysis between the INP number concentrations (at freezing temperatures of -15, -20, and -25 ºC) and meteorological parameters measured in this study. *R* is the correlation coefficient, *P* is the probability value (two tailed), and the sample number is 16.**

| Meteorological parameters | INP number concentrations ($L^{-1}$) | | | | | |
|---|---|---|---|---|---|---|
| | -15 ºC | | -20 ºC | | -25 ºC | |
| | *R* | *P* | *R* | *P* | *R* | *P* |
| Ambient temperature (ºC) | <0.01 | 0.99 | 0.17 | 0.52 | 0.26 | 0.33 |
| Ambient RH (%) | 0.07 | 0.80 | 0.15 | 0.59 | 0.45 | 0.08 |
| Wind direction (degree) | -0.04 | 0.87 | -0.16 | 0.56 | -0.10 | 0.72 |
| Wind speed (km/h) | 0.07 | 0.80 | 0.24 | 0.38 | -0.07 | 0.80 |

[Figure]

**Figure S1. The 10-day HYSPLIT back trajectories for each quartz filter sample. The back trajectories were calculated at the beginning and for every 2h during each 24h sampling period. Each panel represents one sample, and the black stars represent the sampling site. Global Data Assimilation System (GDAS) meteorological data at $1° × 1°$ spatial resolution were used as input to calculate the back trajectories using HYSPLIT.**